# SCOPE-RRG: Symbolic Constraint Preference Optimization for Radiology Report Generation

## Abstract

The complexity of medical image data combined with the variability of natural language generation often leads to inconsistencies, hallucinations, and a lack of clinical grounding, especially in automatically generated radiology reports. To address these challenges, we introduce a task-specific symbolic constraints preference optimization technique, tailored for radiology report generation. A typical radiology report comprises of findings and impression; findings capture the complex visual information from the medical image, for example a chest X-ray, and the impression is the implied conclusion. Our framework leverages on this phenomenon to design clinical rules from existing findings and impressions, that connect the finding and impression as a horn rule. The rules act as an additional, interpretable supervision signal, guiding the preference optimization of Vision–Language Models (VLMs) toward outputs that are fluent as well as clinically coherent. Unlike conventional preference optimization, which relies solely on lexical preferences, our approach enforces alignment with clinically meaningful predicates such as the presence, absence, or severity of key findings. A central feature of this framework is its ability to inject symbolic constraint guidance during optimization, ensuring that generated reports remain both linguistically fluent and clinically coherent. Experimental results on benchmark datasets like MIMIC–CXR-JPG and IU–Xray, demonstrate that our approach substantially improves factual accuracy, and overall report quality compared to zero-shot and standard DPO baselines. We record a significant performance boost across lexical and semantic metrics. These results highlight the promise of clinically interpretable preference optimization as a pathway toward trustworthy radiology report generation in medical AI.

## 1 Introduction

Recent progress in preference optimization has pushed the boundaries on how machines align information across tasks, showing substantial performance in tasks involving image-text pairs Liu et al. (2023). Medical AI tasks have predominantly dealt with image-text pairs Tăuţan et al. (2021); Xia et al. (2024); Tu et al. (2024); Hu et al. (2024). Medical VLMs (Med-VLMs) are being developed that demonstrate striking performance in various use-cases from Medical Visual Question Answering to Radiology Report Generation Zhang et al. (2023); Wu et al. (2023); Moor et al. (2023); Sellergren et al. (2025); Wang et al. (2022b). For image-text alignment across domains, preference optimization has been the most utilized method lately. However, recent research Cho et al. (2025) shows preference optimization, especially Direct Preference Optimization (DPO) does not sufficiently increase the probability of chosen responses Meng et al. (2024). This limitation translates into Med-VLMs in the form of inadequate clinical consistency Zhou et al. (2024); Sun et al. (2024), resulting in Med-VLMs hallucinating. The generated texts appear to be coherent, but are unable to capture the complete corresponding information in medical image. To address this issue, several attempts have been made to propose various preference data curation strategies for better image-text alignment and factual grounding in Med-VLMs Hein et al. (2024); Banerjee et al. (2024); Zhu et al. (2025b); Liu et al. (2025). It is observed, existing preference data curation techniques do not take into account the clinical semantics. As a result, the chosen and rejected pairs used for preference optimization become easily distinguishable as trivial distinguishability leads to generations solely following a lexical template. As a result the model

distribution becomes skewed to learn only the lexical outline and disregarding retention of clinical entities which is essential in a high-stake medical AI system. This skewed distribution increases the risk of misdiagnosis, missing pathology and hallucinated wrong measurements completely misaligned with the X-ray image, rendering the methods non deployable, as the system is expected to be able to generate fluent text which are grounded in clinical semantics. We observe that so far all previous works have treated radiology report generation as a next-token prediction task, without explicitly modeling the clinical reasoning that connects findings to impression. In a real-world setting, a radiology report contains findings and impressions, findings summarizes the observation in the image and the impression is a conclusion to the observation. The journey from the observation to the conclusion remains grounded in standardized reasoning patterns Simpson et al. (2020) that connect visual evidence to diagnostic outcomes. This structured reasoning process ensures both clinical validity and consistency across reports.

Our work anchors on this observation, based on Simpson et al. (2020) we note that a free-text findings and impression can be reduced to a symbolic representation such as, modifier-entity pairs connected via implication. We preprocess the dataset to create horn rules following this structure and use it as a symbolic constraint for radiology report generation. We first extract entities and the corresponding modifiers from the findings and impressions of a report. A predicate rule is created based on the extracted modifier-entity pairs. Following this we train a neural verifier, that checks whether a rule and the corresponding report are aligned or not. The preference dataset is then curated by performing multiple sampling of the medical-VLM and ranking the outputs using the verifier and choosing the two top-most ranked outputs as chosen and rejected pairs, thereby removing trivial distinguishability. This curation strategy embeds the clinical semantics in the preference dataset. Importantly, the rule creation and the data curation strategy is agnostic to the underlying vision-language framework and only assumes access to domain specific medical text guidelines. Our preference data curation technique along with the designed symbolic rules mitigate the issue of generated reports being lexical sound but clinically incorrect. With the aforementioned resources in our hand we design an optimization framework that utilizes the symbolic clinical rules (approved by practicing clinicians) as interpretable supervision, along with the preference data. We thus, put forth a joint optimization method that fine-tunes VLMs to generate fluent and clinically coherent outputs.

Our contributions are:

1. Symbolic constraint–guided preference data construction: We propose SCOPE-RRG, for constructing preference datasets guided by symbolic clinical constraints, enabling the generation of supervision signals that reflect clinically meaningful relationships rather than purely surface-level differences.

2. Neural rule verifier for logical consistency: We introduce a neural verifier that evaluates the consistency between generated outputs and structured rules, providing interpretable, rule-based assessments of logical and clinical validity.

3. Symbolic constraint–guided joint preference optimization: We develop a joint preference optimization framework that incorporates symbolic constraints during training, encouraging models to produce outputs that are both fluent and aligned with structured domain knowledge.

## 2 Background

### 2.1 Radiology Report Generation via Medical Vision-Language Models

Medical vision-language models (VLMs) leverage transformer-based architectures and multimodal pretraining on medical images and reports for tasks such as radiology report generation and visual question answering (Wang et al., 2022b; Sellergren et al., 2025; Moor et al., 2023). Radiology report generation commonly uses encoder-decoder architectures (Vinyals et al., 2014; Xu et al., 2015; Pan et al., 2020), with improvements including image-text matching, hierarchical LSTMs, and memory modules for long-text decoding (Wang et al., 2021; Chen et al., 2020; Wang et al., 2022a). To mitigate data bias, some approaches integrate external knowledge via knowledge graphs, using graph neural networks, dynamic updates, or knowledge distillation (Li et al., 2019; 2023; Huang et al., 2023; Liu et al., 2021; Zhang et al., 2020; Kale et al., 2023).

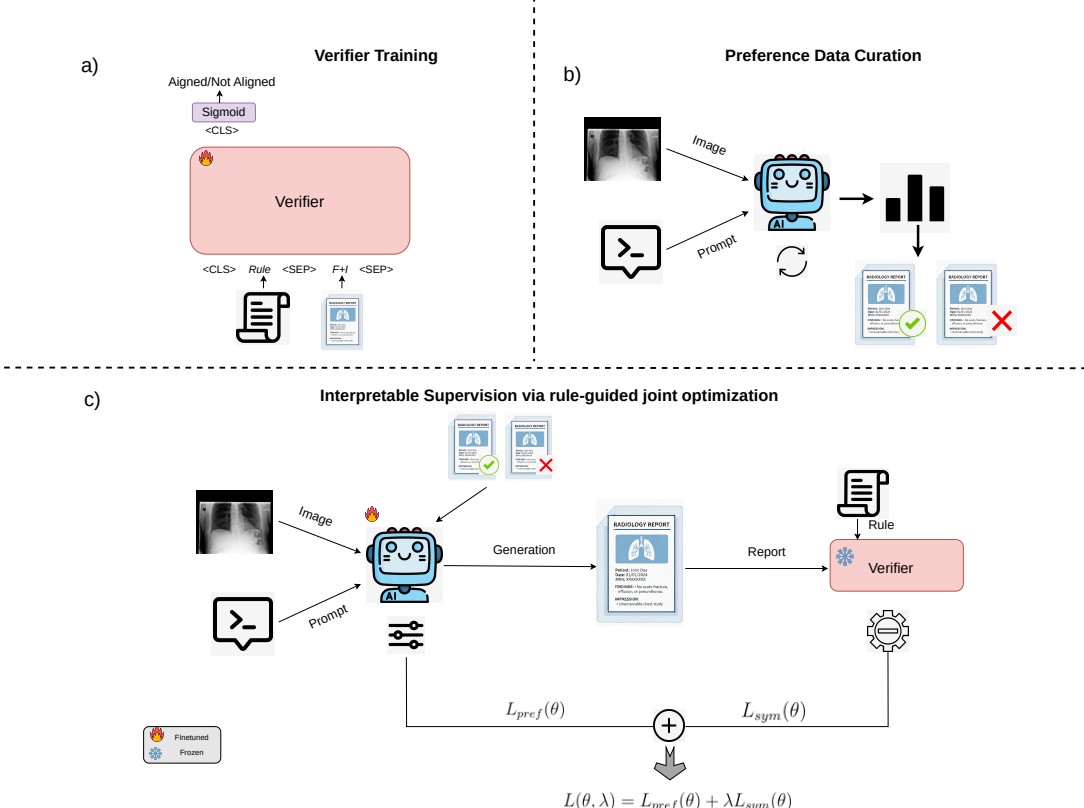

Figure 1: (a) Neural Verifier is trained to verify a radiology report with a symbolic constraint (b) Using the verifier we create the preference data (c) At each step the joint optimization process inputs a chest x-ray and a prompt to the VLM that produces the report which is verified and the verifier gives a score between 0-1. Finally, the proposed loss function optimizes the VLM to generate fluent and clinically grounded radiology reports.

## 2.2 Preference optimization for medical tasks

Recent work has explored extending preference optimization to better capture domain-specific requirements in medical and multimodal settings. RRG-DPO Liu et al. (2025) propose a clinically relevant and abnormal-aware preference data curation approach. This method proposes a preference data curation to tackle the problem generating conflicting abnormality sentences. Similarly, MMedPO Zhu et al. (2025a) introduces a novel framework that curates multimodal preference data through two specialized strategies: injecting plausible hallucinations using target Med-LVLMs or GPT-4o, and provoking lesion region neglect via local lesion-noising tools. Furthermore, the method quantifies the importance of these samples by assigning clinical relevance weights derived from a collaborative multi-agent Med-LLM debating process and confidence scores from visual lesion detection tools. This approach significantly enhances factual accuracy and visual understanding, outperforming existing baselines across diverse medical tasks such as visual question answering and radiology report generation. We closely observe that the recent methods assume the ground truth as preferred response. However, for preference optimization to be effective, the underlying pre-trained model must be capable of generating at least one output that can be considered a preferred sample Rafailov et al. (2024). Therefore, we build our framework on this principle

While these methods heavily rely on Direct Preference Optimization (DPO) whereas, SimPO (Simple Preference Optimization) Meng et al. (2024) proposes a simplified alternative that removes the need for a reference policy. SimPO directly optimizes the margin between preferred and dispreferred responses under the current

model, reducing computational overhead and simplifying training. Following, this we propose a modified objective where the verifier acts as a margin. Our proposed method directly optimizes the VLM framework by using task-specific symbolic constraints as a supervision signal along with the SCOPE-RRG objective.

## 3 Methodology

Our proposed method SCOPE-RRG, as shown in Fig. 1, draws inspiration from the trustworthy alignment line of research Dai et al. (2024); Liu et al. (2024); Asadi et al. (2025). All of the previous methods explore how meticulously curated preference datasets can be useful for steering a model to human aligned and semantically coherent outputs. However, none of the previous methods try to incorporate explicit symbolic rule for constraining the preference optimization. In a clinical setting, reliance on explicit clinical rules gains more weight-age over only human curated preference datasets. Recent VLM based works in medical NLP, particularly in radiology report generation, concentrate on maximizing fluency, coherency and capturing the reporting style Hein et al. (2024); Banerjee et al. (2024); Zhu et al. (2025b). This frontier of research has led to generation of fluent and clinically styled text. However, in a sensitive domain like medical NLP, fluency alone do not suffice to accept and trust generated outputs. A degree of factual grounding in symbolic clinical knowledge is required, which in turn raises acceptability and increases trust. Nonetheless, all previous medical VLMs missout on this perspective. We aim to address this crucial aspect via finetuning of medical VLMs with symbolic clinical rules. Therefore, our finetuning method enable the VLMs to generated fluent as well as clinically grounded radiology reports.

### 3.1 Data Preprocessing

A radiology report consists of findings and impressions. An example of findings and impressions is given below.

> **Findings**: The cardiac silhouette is enlarged, consistent with cardiomegaly. Lung volumes are stable and remain low. No evidence of pneumothorax is seen. Minimal blunting of the right costophrenic angle is noted. No focal infiltrates are identified.
> **Impressions**: cardiomegaly with mild right pleural effusion.

We observe that each free-text radiology reports can be reduced to a rule structure, such as $findings \implies impression$. Findings, in this underlying structure comprises the exact clinical knowledge from the image, and impression contains the conclusion. This overall structure represents the manner in which a report is written in a real-world setting.

> cardiomegaly and low lung volume and no pneumothorax and minimal right costphrenic blunting and no focal infiltrates
>
> **implies**
>
> cardiomegaly and mild right pleural effusion

Therefore, we include a pre-processing stage to reduce a free-text radiology into a natural language clinical rule. We connect each finding and impression via an implication, in the form of a predicate rule. Before training, we extract natural language predicate rules from each of the data samples. Following this stage, each dataset instance has a X-ray image, report and the corresponding natural language predicate rule connecting the findings and impression via an implication. The preprocessing pipeline consists of the following steps:

### 3.1.1 Entity Extraction

Each report is parsed to identify radiological entities.

> **Entities from Findings**: cardiomegaly, lung volume, pneumothorax, right costphrenic, infiltrates
>
> **Entities from Impressions**: cardiomegaly, pleural effusion

To obtain structured clinical rules from free-text radiology reports, we employ the Stanza natural language processing toolkit Qi et al. (2020). Specifically, we use the MIMIC-trained English pipeline with the i2b2 Named Entity Recognition (NER) processor.

These extracted entities serve as the building block for the natural language predicate rule. Following, this we pair this with modifiers that signal the intensity of the disease in the provided chest x-ray. The extracted entities, clubbed with the modifiers, build the final predicate rule. The reduction of the unstructured radiology reports into structured rules, builds ground for grouding radiology report generation with domain-specific constraints.

### 3.1.2 Modifier Detection

Once entities are extracted, we perform contextual analysis to capture descriptive information that inform us about the instensity of the disease. These contextual cues are referred to as **modifiers**.

> **Modifiers from Findings**: low, no, minimal
>
> **Modifiers from Impressions**: mild

Entities along with modifiers form entity-modifier pairs that are used as individual literals in the final rule.

> **Entity modifier pair from Findings**:cardiomegaly, low lung volume, no pneumothorax, minimal right costrophrenic infiltrates
>
> **Entity modifier pair from Impressions**: cardiomegaly and mild pleural effusion

To identify these modifiers we first do a contextual analysis of the dataset and create a set of modifiers. With each radiology report we first extract the entity and search within a window of ten forward and backward tokens to find modifiers that map to our set of modifiers. By combining these approaches, the algorithm 2 produces structured entity–modifier pairs, which are later concatenated to form a predicate rule in natural language.

### 3.1.3 Horn Rules Formation

Following entity extraction and modifier detection, each entity–modifier pair forms the individual literal of the final rule. Therefore, the final horn rule captures the underlying nuanced clinical reasoning required for generating a fluent and clinically grounded radiology report. Throughout this process we make sure our method adheres to natural language rules.

Finally, a natural language Horn rule is formed that takes the following structure:

$$p_1 \wedge p_2 \wedge \cdots \wedge p_k \;\; \rightarrow \;\; q,$$

> cardiomegaly $\wedge$ low lung volume $\wedge$ no pneumothorax $\wedge$ minimal right costrophrenic infiltrates $\rightarrow$ cardiomegaly $\wedge$ mild pleural effusion

where the conjunction of predicates $(p_1, p_2, \ldots, p_k)$ represents evidence extracted from the findings of the radiology report, and $q$ denotes a entity-modifier pair from the impression. The entity-modifier pairs capture the disease and the intensity and represent it in the conjunction form, thus we chose the horn rule representation.

By systematically composing entity-level predicates into natural language Horn rule, we create an interpretable connection between raw textual descriptions and clinically relevant rules. This logical representation facilitates downstream reasoning, supports clinically grounded guidance of VLMs, which we further incorporate in our downstream task of radiology report generation.

## 3.2 Clinical Rule guided verifier

Our methodology for developing a verifier model is primarily inspired by the work of Clark et al. (2021), which established that transformer models can be trained to effectively reason over rules and paragraphs expressed in natural language.

### 3.2.1 Neural Verifier

We adopt this approach and implement the verifier using a *RoBERTa-large* architecture, framing the verification task as a binary classification problem. Given a radiology report $R$ (comprising Findings $F$ and Impression $I$) and a Horn rule $h$, the model predicts whether the report is *Aligned* or *Not Aligned* with the rule. To train the model, we construct a large-scale dataset as described earlier, labeling aligned samples as 1 and misaligned samples as 0. The report and rule are concatenated into a structured input sequence $X = \texttt{<CLS>}\ h\ \texttt{[SEP]}\ R\ \texttt{[SEP]}$, which is tokenized and passed through RoBERTa to obtain a contextual embedding $z = \text{RoBERTa}_\theta(X) \in \mathbb{R}^d$, corresponding to the pooled representation of the $\texttt{<CLS>}$ token. This representation summarizes the entire input sequence and is commonly used for downstream classification tasks in transformer-based models. A linear classification head then projects this embedding into a scalar logit $\ell = Wz + b$, which is mapped through a sigmoid function to obtain the alignment probability $\hat{y} = \sigma(\ell) = P_\theta(y = 1 | R, h) \in [0, 1]$. We keep all symbolic representations in fluent natural language, making the rules easily interpretable and enabling a transition from symbolic rule verification to neural verification. The RoBERTa model is fine-tuned using a binary cross-entropy (BCE) loss, with accuracy serving as the primary evaluation metric due to the balanced class distribution in the dataset.

---

**Algorithm 1:** Interpretable Supervision via symbolic-constraint guided preference optimization

---

**Input:** $\mathcal{D} = \{x_v^{(i)}, x_t^{(i)}, r^{(i)}\}_{i=1}^N$: Dataset; $x_p$: Prompt; $\pi_\theta(\cdot, \cdot)$: Med-VLM; $\mathcal{V}(\cdot, \cdot)$: Neural Verifier
**Output:** Optimized model parameters $\theta^\star$ for Med-VLM.
Initialize empty preference dataset $\mathcal{D}_P$
**foreach** $(x_v, x_t) \in \mathcal{D}$ **do**
    Generate multiple responses: $r_m \leftarrow \pi_\theta(x_v, x_p)$ ;         // *Generate candidate reports*
    **foreach** $r_i \in r_m$ **do**
        Compute logical consistency: $v_i \leftarrow \mathcal{V}(r_i, x_t)$ ;      // *Compute verifier score*
    Preferred Response: ;          // *Choose top two ranked responses*
    $y_w \leftarrow$ Highest Verifier Score
    $y_l \leftarrow$ Second highest Verifier Score
    Add $(x_v, x_t, y_w, y_l, r)$ to $\mathcal{D}_P$ ;      // *Collect preference and verifier info*
**foreach** $(x_v, x_t, y_w, y_l, r) \in \mathcal{D}_P$ **do**
    Compute preference loss: $L_{\text{pref}} \leftarrow -\log \sigma\big(\pi_\theta(x, y_w) - \pi_\theta(x, y_l)\big)$
    Compute symbolic loss: $L_{\text{sym}} \leftarrow |1 - v|$
    Compute $\lambda$: $\lambda \leftarrow \text{softplus}(\lambda_{\text{param}})$
    Compute joint loss: $L_{joint} \leftarrow L_{\text{pref}} + \lambda L_{\text{sym}}$ ;      // *SCOPE-RRG*
    Backpropagate $L_{\text{joint}}$ and update $\theta$
    Backpropagate $-\lambda L_{sym}$ and update $\lambda$
**return** $\theta^\star \leftarrow$ *optimized Med-VLM parameters*

---

### 3.2.2 Neural Verifier Training

We prepare a dataset with aligned and misaligned samples based on the aforementioned rule extraction strategy. The details of aligned/misaligned data preparation is mentioned in Appendix A.2.1. Following, this the neural verifier is trained to classify whether a given report-rule pair is aligned/misaligned. We use the following loss function to train the neural verifier: **Neural Verifier Loss ($L_V$):** The neural verifier is trained to predict whether a radiology report $r$ is logically consistent with a rule $R$. We use a binary cross-entropy (BCE) loss, which penalizes the verifier when its predicted probability diverges from the ground-truth label $y \in \{0, 1\}$:

$$L_V = \mathbb{E}_{(r,R,y) \sim D_{rule}} \left[ y \cdot \log p(y = 1 \mid r, R) + (1 - y) \cdot \log p(y = 0 \mid r, R) \right] \tag{1}$$

This objective ensures that the verifier learns to assign high confidence to logically consistent report–rule pairs while suppressing inconsistent ones. Here, $D_{rule}$ is the verification dataset, and $y$ is the ground-truth label for a given report $r$ and rule $R$. The verifier is first trained via this loss mentioned in 1.

### 3.3 Interpretable Supervision via symbolic-constraint guided preference optimization

In clinical report generation, stylistic fluency alone is insufficient, factual correctness and adherence to clinical rules are equally essential for producing reliable, clinically grounded outputs. To enforce clinical consistency during the fine-tuning of the vision–language model (VLM), we propose SCOPE-RRG, a symbolic-constraint guided preference optimization framework. We introduce interpretable supervision via a clinical rule verifier that evaluates whether generated reports satisfy domain-specific constraints.

The training process relies on a preference dataset $\mathcal{D}_P$, consisting of triplets of the form $(R, y_w, y_l)$. For a given clinical rule $R$, $y_w$ denotes the preferred ("winning") report and $y_l$ represents the dispreferred ("losing") report. The SCOPE-RRG objective encourages the model to increase the likelihood margin between the preferred and dispreferred reports, thereby aligning the model with clinically appropriate reporting style.

To ensure that the report generation model remains both linguistically fluent and clinically grounded, we formulate a joint optimization problem that integrates preference learning with rule verification. Specifically, the objective minimizes the SCOPE-RRG preference loss while encouraging the verifier output to approach 1, which indicates strong consistency with the corresponding clinical rules. This leads to the following constrained optimization problem:

$$\theta^\star = \arg \min_\theta \mathbb{E}_{(x,y_w,y_l) \sim \mathcal{D}_P} \left[ -\log \sigma \left( \beta \log \pi_\theta(y_w|x) - \beta \log \pi_\theta(y_l|x) \right) \right] \tag{2}$$

While this objective aligns the model with preference signals, preference supervision alone can encourage shortcut learning and overly templated generations when preferred and rejected pairs are easily separable. To mitigate this, we introduce a neural verifier that provides symbolic feedback based on domain-specific clinical rules. For a generated report $y_e$ and rule r, the verifier predicts a rule consistency score.

$$Verifier(y_e, r) \in [0, 1], \tag{3}$$

where higher score indicates stronger rule satisfaction. This induces a symbolic regularization loss.

$$L_{sym}(\theta) = \mathbb{E}_{(x,y_e) \sim \mathcal{D}_P} \left[ 1 - Verifier(y_e, r) \right]. \tag{4}$$

We utilise the interpretable supervision signal as a soft feedback signal that shapes the preference optimization landscape. The overall SCOPE-RRG objective is therefore defined as:

$$L(\theta, \lambda) = L_{pref}(\theta) + \lambda L_{sym}(\theta) \tag{5}$$

where $\lambda \geq 0$ controls the trade-off between preference alignment and clinical consistency. The first term promotes fluent and preference-aligned report generation, while the second discourages generations that violate learned clinical rule structure. Consequently, the generator is updated using the combined gradient

$$\nabla_\theta L = \nabla L_{pref}(\theta) + \lambda \nabla L_{sym}(\theta) \tag{6}$$

allowing symbolic feedback to directly modify the optimization trajectory of the generator. Rather than fixing $\lambda$, we update it dynamically during training using a primal-dual inspired strategy. When rule violations increase, the symbolic feedback term is strengthened, forcing optimization to prioritize clinical consistency; when violations decrease, optimization places relatively greater emphasis on preference alignment. This adaptive mechanism provides a principled balance between linguistic fluency and factual correctness.

Thus, SCOPE-RRG can be interpreted as constraint-guided reward shaping for preference optimization, where symbolic feedback counteracts shortcut preference signals and steers generation toward clinically grounded outputs. This adaptive mechanism provides a principled way to balance stylistic fluency and clinical correctness. Algorithm 1 summarizes the training workflow, and the corresponding gradient updates are provided in Appendix B.

## 4 Experimental Setup

### 4.1 Dataset

We evaluate SCOPE-RRG on two widely used public radiology benchmarks: MIMIC-CXR-JPG Johnson et al. (2019) and IU-Xray Demner-Fushman et al. (2015). MIMIC-CXR-JPG consists of approximately 377,000 frontal and lateral chest X–ray images paired with free-text radiology reports, providing a large-scale setting for multimodal learning. IU-Xray contains around 7,000 studies, each with corresponding chest X–ray images and reports, and is commonly used to evaluate generalization and cross-dataset performance.

For both datasets, we follow a consistent preprocessing pipeline as aforementioned. The datasets are split into training, validation, and test sets following the recommended splits, ensuring no patient overlap across sets. This standardized preprocessing ensures that both textual and visual modalities are compatible with our multimodal model, allowing for robust evaluation of linguistic quality and clinical consistency in generated reports. The dataset preprocessing details along with splits are provided in Appendix A.1. We also provide the experimental setup in Appendix B.1.

## 5 Results

We evaluate SCOPE-RRG across diverse settings, encompassing both in-task and cross-task scenarios. Broadly, our evaluation framework captures consistent performance trends as well as fine-grained, clinically grounded behavior. We observe that SCOPE-RRG delivers consistent improvements across both dimensions. Positive performance gains across lexical and semantic metrics shows the generalisability of our proposed approach across in-task and cross-task pretrained models.

### 5.1 Global Assessment

In this section we present comparison of SCOPE-RRG against existing DPO baselines and zero-shot settings, for both lexical and semantic metrics. We observe a positive increment across majority of lexical and semantic metrics.

#### 5.1.1 Lexical Analysis

We evaluate SCOPE-RRG on traditional lexical metrics like BLEU Papineni et al. (2002), Rouge-Score Lin (2004) scores. In Table 1, the metrics provide a convenient means of measuring word overlap and syntactic similarity between generated and reference texts. Our method, SCOPE-RRG demonstrates a considerable increase in lexical scores. The main benefit of SCOPE-RRG is not merely higher lexical overlap, but better preservation of clinically relevant content while maintaining strong report quality. This distinction is important because lexical metrics can reward templated outputs that resemble references at the surface level while overlooking omissions of critical medical entities.

In contrast, vanilla DPO often improves lexical alignment by favoring common reporting patterns and recurrent templates, which can yield fluent outputs but may miss clinically salient findings. SCOPE-RRG

| Models | IUX | | | | | MIMIC-CXR-JPG | | | | |
|---|---|---|---|---|---|---|---|---|---|---|
| | BL-1 | BL-2 | BL-3 | BL-4 | RG-L | BL-1 | BL-2 | BL-3 | BL-4 | RG-L |
| Med-Gemma (ZS) | 27.39 | 14.72 | 07.19 | 02.33 | 24.17 | 15.17 | 7.66 | 3.64 | 1.50 | 18.53 |
| + DPO | 25.17 | 14.27 | 07.48 | 02.73 | 24.09 | 25.45 | 12.60 | 06.02 | 02.13 | 16.55 |
| + RRG-DPO | 21.89 | 09.52 | 03.51 | 01.22 | 19.54 | 26.69 | 11.38 | 06.16 | 02.64 | 18.89 |
| + MMedPO | 26.40 | 13.58 | 06.26 | 02.54 | 23.63 | 21.56 | 10.58 | 05.09 | 02.33 | 18.57 |
| +SCOPE-RRG (Ours) | 29.36 | 17.48 | 09.97 | 05.39 | 27.71 | 22.90 | 13.62 | 05.66 | 03.55 | 19.28 |
| Med-Flamingo (ZS) | 17.55 | 08.44 | 01.27 | 0.11 | 15.36 | 12.49 | 05.75 | 01.93 | 00.33 | 15.64 |
| +DPO | 17.83 | 08.39 | 02.53 | 00.69 | 18.77 | 20.04 | 07.77 | 03.31 | 00.81 | 16.77 |
| +RRG-DPO | 16.88 | 07.42 | 02.11 | 00.23 | 17.67 | 17.04 | 06.77 | 02.83 | 00.52 | 17.11 |
| +MMedPO | 17.23 | 08.19 | 02.63 | 00.43 | 18.62 | 19.04 | 08.17 | 03.11 | 00.68 | 17.97 |
| +SCOPE-RRG (Ours) | 18.41 | 09.35 | 03.06 | 00.55 | 19.22 | 22.86 | 09.91 | 01.87 | 00.45 | 18.45 |
| LLaVa-Rad (ZS) | 23.44 | 10.85 | 04.68 | 01.94 | 24.90 | 27.53 | 16.61 | 10.53 | 06.64 | 23.54 |
| +DPO | 23.51 | 10.96 | 04.75 | 01.97 | 24.96 | 27.53 | 16.64 | 10.52 | 06.61 | 23.61 |
| +RRG-DPO | 23.39 | 10.76 | 04.72 | 02.02 | 25.03 | 27.47 | 16.63 | 10.55 | 06.66 | 23.62 |
| +MMedPO | 23.32 | 10.94 | 04.66 | 01.87 | 24.69 | 27.19 | 16.42 | 10.41 | 06.55 | 23.41 |
| +SCOPE-RRG (Ours) | 25.13 | 12.13 | 05.52 | 02.48 | 24.02 | 29.25 | 17.25 | 12.58 | 07.45 | 26.45 |

Table 1: Comparison of VLMs for radiology report generation across multiple evaluation metrics, including BLEU, and ROUGE score for assessing the lexical accuracy. The table highlights the improvements achieved by our SCOPE-RRG framework over zero-shot and standard DPO baselines.

mitigates this by balancing preference alignment with symbolic verifier feedback, encouraging both natural report structure and faithful retention of clinical entities. To better understand we observe a qualitative example.

1. *Ground Truth*: *findgings:findings include normal heart size and mediastinal contours, **stable tortuosity of the thoracic aorta**, and absence of consolidation, pleural effusion, or pneumothorax. impression: no acute cardiopulmonary abnormality.*

2. *DPO*: *findings: heart size, mediastinal and hilar contours are normal. lungs and pleural surfaces are clear. impression: no acute cardiopulmonary abnormality*

3. *SCOPE-RRG*: *findings: heart size, mediastinal and hilar contours are normal. **the aorta is tortuous**. lungs are clear. no pleural effusion or pneumothorax. impression: no acute cardiopulmonary process.*

We see the DPO generation follows a general template which is similar to the ground truth on the lexical level. On close observation we see the DPO generation misses on key anatomical entity such as **tortusity of aorta**, which is clearly captured in SCOPE-RRG generation. Thus, it demonstrates the utility of the symbolic verifier and the supervision it provides during the preference optimization setup. The highlighted scores show the efficacy of our method.

### 5.1.2 Semantic Analysis

Following the lexical evaluation metrics we move on to semantic evaluation metrics, like ClinicalBERTScore Shor et al. (2023), RadGraph-F1 Jain et al. (2021), GREEN Ostmeier et al. (2024). These metrics are particularly important for radiology reporting, where subtle differences such as "no pleural effusion" versus "pleural effusion present" can fundamentally alter diagnostic meaning, maintaining similar lexical structure. Across datasets, SCOPE-RRG consistently improves over both zero-shot and DPO baselines on semantic

| Models | IUX | | | MIMIC-CXR-JPG | | |
|---|---|---|---|---|---|---|
| | CBS | RG-F1 | GREEN | CBS | RG-F1 | GREEN |
| Med-Gemma (ZS) | 87.69 | 26.02 | 00.40 | 87.12 | 16.17 | 00.20 |
| + DPO | 84.77 | 22.49 | 00.38 | 86.51 | 19.99 | 00.22 |
| + RRG-DPO | 85.07 | 18.53 | 00.32 | 89.00 | 18.27 | 00.22 |
| + MMedPO | 85.34 | 25.48 | 00.42 | 90.03 | 16.80 | 00.22 |
| +SCOPE-RRG (Ours) | 88.17 | 31.00 | 00.48 | 90.39 | 18.18 | 00.25 |
| Med-Flamingo (ZS) | 86.44 | 16.64 | 00.38 | 80.55 | 08.45 | 00.23 |
| +DPO | 88.35 | 19.22 | 00.42 | 86.61 | 10.47 | 00.27 |
| + RRG-DPO | 87.68 | 18.00 | 00.39 | 86.79 | 11.55 | 00.23 |
| + MMedPO | 85.68 | 16.89 | 00.45 | 87.79 | 15.55 | 00.26 |
| +SCOPE-RRG (Ours) | 89.55 | 20.26 | 00.51 | 88.45 | 13.72 | 00.31 |
| LlaVa-Rad (ZS) | 90.08 | 24.29 | 00.45 | 90.01 | 24.29 | 00.37 |
| +DPO | 83.42 | 25.39 | 00.47 | 90.11 | 24.32 | 00.38 |
| +RRG-DPO | 83.40 | 25.56 | 00.46 | 90.09 | 24.40 | 00.38 |
| +MMedPO | 83.42 | 25.26 | 00.47 | 90.06 | 24.27 | 00.40 |
| +SCOPE-RRG (Ours) | 84.41 | 26.55 | 00.48 | 89.88 | 24.55 | 00.44 |

Table 2: Comparison of VLMs for radiology report generation across multiple evaluation metrics, including ClinicalBERTScore (CBS), GREEN and RadGraph-F1 (RG-F1) for assessing the accuracy of clinically relevant relational structures. The table highlights the improvements achieved by our SCOPE-RRG framework over zero-shot and standard DPO baselines.

metrics, demonstrating stronger alignment with clinically relevant findings and more faithful report generation. Notably, the gains in semantic performance are more pronounced than those observed in lexical overlap metrics, suggesting that the primary benefit of symbolic constraint guidance lies in improving clinical correctness rather than merely surface-level phrasing. Preference optimization can favor lexically fluent yet templatized outputs that satisfy preference datasets due to trivial distinguishability. However, this leads to omition of important clinical entities or the relational structure between findings and impression. In contrast to that SCOPE-RRG, encourages preservation of lexical structure as well as the clinical entities consistent with symbolic rules. RadGraph-based improvements establishes that SCOPE-RRG generated lexically fluent and clinically coherent reports, where higher entity and relation overlap indicates that symbolic constraints help the model retain medically salient concepts that may otherwise be lost under trivially distinguishable preference datasets. These gains suggest that the verifier does not simply improve report style, but shape the model distribution towards balanced report generation.

Appendix A.3 shows the failure cases as well. Cumulatively, lexical and semantic results demonstrate that while large vision-language models can achieve strong template coverage, their relational reasoning remains limited. Preference Optimization helps address this gap but lacks clinical grounding and entity preservation when applied alone. Therefore, in SCOPE-RRG, integration of a symbolic verifier into the optimization loop strikes a better balance, allowing the model to preserve linguistic coherence as well as keeping the generation clinically grounded.

## 5.2 Nuanced Assessment

SCOPE-RRG improves Med-VLMs by enforcing clinically coherent generation through symbolic-constraint preference optimization. It yields consistent in-task gains and improved cross-task generalization, indicating better alignment with both training distributions and underlying clinical structure. A neural verifier evaluates outputs for clinical consistency, and its signals are incorporated during optimization to reinforce correct behaviors and suppress hallucinations. This leads to outputs that are not only fluent but also factually accurate and rule-compliant. Overall, SCOPE-RRG enables more reliable and clinically grounded report generation, improving the robustness of Med-VLMs for real-world use.

We evaluate SCOPE-RRG across both in-task pretrained Med-VLMs (Med-Gemma, LLaVA-Rad) and a cross-task pretrained model (Med-Flamingo). Across all models, symbolic verifier-guided preference optimization improves clinical consistency while preserving fluent report generation. Notably, SCOPE-RRG generalizes beyond the pretraining distribution, helping even cross-task models transfer effectively to radiology report generation. These results suggest that task-specific symbolic constraints provide a robust, model-agnostic supervision signal for steering diverse Med-VLMs toward semantically faithful outputs

### 5.2.1 Cross-Institution generalization of the neural verifier

We assess the cross-institution generalization of the MIMIC-trained verifier by evaluating it on the IUX dataset and IUX-trained verifier by evaluating on MIMIC. Interestingly, very similar performance is maintained, indicating that the learned clinical rule semantics transfer

| Verifier | B-1 | B-2 | B-3 | B-4 | RG-L |
|---|---|---|---|---|---|
| Verifier trained on MIMIC+ evaluated on MIMIC (within institution) | 22.90 | 11.38 | 5.66 | 2.64 | 18.89 |
| Verifier trained on MIMIC+evaluated on IUX (outside institution) | 24.22 | 13.69 | 7.08 | 3.45 | 23.05 |
| Verifier trained on IUX+evaluated on IUX (within institution) | 29.36 | 17.48 | 9.97 | 5.39 | 27.71 |
| Verifier trained on IUX+evaluated on MIMIC (outside institution) | 30.77 | 18.17 | 11.38 | 5.99 | 21.82 |

Table 3: Cross-institution generalization results of neural verifiers trained on MIMIC and IU-Xray, demonstrating robust transfer of learned clinical rule semantics across datasets despite domain shifts in reporting style and pathology

robustly across institutions despite differences in reporting conventions, stylistic phrasing, and pathology distributions. This suggests that the verifier captures rule-level relational structure rather than overfitting to institution-specific wording. Minor fluctuations that do occur are attributable to natural domain shift, but overall the results demonstrate strong robustness of both the verifiers when applied to unseen institutions and reporting styles. This concludes that our neural verifier training strategy is generalizable and robust across chest-Xray datasets.

### 5.2.2 Verifier Sensitivity and Threshold Calibration

We evaluate six settings reflecting increasing constraint strength: *Med-Gemma (Verifier accuracy = 10, 40, 50, 70, 90 )*, and our full *SCOPE-RRG* model, which directly incorporates verifier feedback into the optimization. As shown in Table 4, improving verifier strength from 10 to 90 yields positive gains across metrics (e.g., BLEU-4 im-

| Verifier | B-1 | B-2 | B-3 | B-4 | RG-L | RG-F1 | GREEN |
|---|---|---|---|---|---|---|---|
| SCOPE-RRG, Ver10% | 10.12 | 7.69 | 2.08 | 0.45 | 10.05 | 6.41 | 00.10 |
| SCOPE-RRG, Ver40% | 19.11 | 11.78 | 7.36 | 3.33 | 20.11 | 10.28 | 00.23 |
| SCOPE-RRG, Ver 50% | 23.84 | 13.62 | 7.32 | 3.69 | 22.65 | 27.80 | 0.25 |
| SCOPE-RRG, Ver 70% | 24.43 | 13.70 | 6.83 | 3.21 | 22.96 | 26.79 | 0.31 |
| SCOPE-RRG, Ver 90% | 26.01 | 15.13 | 8.23 | 4.24 | 24.85 | 29.41 | 0.38 |
| SCOPE-RRG, Ver98% (Reported) | 29.36 | 17.48 | 9.97 | 5.39 | 27.71 | 31.00 | 00.48 |

Table 4: Ablation on verifier sensitivity demonstrating that increasing verifier strength improves both fluency and clinical consistency, validating the role of symbolic constraint guidance in preference optimization

proves from 0.45 → 4.24,
and RG-L from 10.05 →
24.85), indicating the framework is highly sensitive to verifier accuracy. Overall, the empirical scores demonstrate that the symbolic constraint influences fluency and clinical consistency in a consistent manner. These results suggest that our proposed optimization framework effectively balances linguistic quality with clinically grounded reasoning.

### 5.2.3  Qualitative Evaluation

| Evaluator | Fluency | Coherency |
|---|---|---|
| Radiologist | 4.19±1.15 | 4.35±1.08 |
| DeepSeek-R1-Distill-Qwen-14B | 4.33±0.76 | 3.92±1.10 |
| Qwen2.5-72B-Instruct | 4.02±0.88 | 3.85±0.98 |

Table 5: Expert and LLM-based evaluation of SCOPE-RRG on fluency and coherency (mean±std).

**Human and LLM-Based Evaluation.** While expert radiologist evaluation remains the gold standard for assessing clinical report quality, it is costly and difficult to scale. To complement expert assessment, we employ an LLM-based evaluation protocol in which strong reasoning models assess generated reports against references using a 1–5 Likert scale (prompt details in Appendix A.2).

Evaluation is conducted along two axes, **Fluency Coherency** as shown in Appendix A.2 . The radiologist are instructed to follow the same instruction as A.2 and score each sample on each axes on a likert scale of 1-5.

To further validate these reports, we conduct a targeted expert evaluation with a radiology collaborator on 50 samples balanced across normal and abnormal studies. Reports are rated using the same fluency and coherency criteria. As shown in Table 5, expert assessment aligns closely with LLM-based judgments, with SCOPE-RRG receiving strong scores across both dimensions.

Notably, improvements in coherency are particularly significant, as they suggest that the gains observed in automatic semantic metrics translate into clinically meaningful improvements in factual correctness and consistency. Following, this to ensure the improvements in coherency we also perform a inter-annotator agreement for the LLM evaluators. We observed a Cohen kappa score of 0.67 for coherency. This indicates the consistency of fluency as well as coherency in our generations.

Overall, these results provide complementary evidence that SCOPE-RRG improves not only automatic lexical and semantic metrics, but also human-perceived report quality, strengthening the case that symbolic verifier supervision yields clinically relevant benefits beyond what is captured by automated metrics alone.

## 6  Conclusion

In this work, we introduced a clinically interpretable, rule-guided extension of Direct Preference Optimization for radiology report generation. By leveraging the inherent structure of radiology reports—specifically the relationship between findings and impressions—we formulate Horn-rule–based constraints that explicitly connect visual evidence with clinical conclusions. These structured constraints are integrated into training through a neural verifier, which evaluates rule consistency and provides targeted supervision signals. This allows the model to balance linguistic fluency with clinical faithfulness, reducing reliance on purely surface-level patterns.

Extensive experiments on MIMIC–CXR-JPG and IU–Xray demonstrate consistent improvements across both lexical and clinically grounded metrics, along with better alignment between generated findings and impressions. Qualitative and human evaluations further indicate reduced hallucinations and improved preservation of clinically relevant details. Notably, the gains are achieved without sacrificing fluency, suggesting that constraint-guided optimization can enhance reasoning while maintaining natural language quality.

Overall, our results highlight that incorporating symbolic clinical constraints into preference optimization offers a practical and interpretable pathway for improving the reliability of medical vision–language models. This approach moves beyond purely data-driven alignment toward structured, clinically grounded generation, bringing such systems closer to safe and trustworthy real-world deployment.

**Broader Impact Statement**

This work advances the development of more reliable and clinically grounded medical vision–language models by integrating interpretable, rule-guided constraints into preference optimization. By explicitly modeling the relationship between findings and impressions, the proposed framework improves consistency, reduces hallucinations, and enhances the factual quality of generated reports. Such improvements are particularly important in high-stakes medical settings, where even minor inconsistencies can impact clinical interpretation.

The interpretability of the approach, through explicit rules and a verifier mechanism, provides an additional layer of transparency compared to purely data-driven methods. This can help build trust among clinicians and facilitate better understanding of model behavior. In practical settings, systems built on such framework could serve as assistive tools to support radiologists by generating more structured and clinically coherent reports, potentially improving efficiency and standardization in reporting workflows.

More broadly, this work highlights the value of combining symbolic knowledge with modern learning-based approaches, offering a pathway toward AI systems that are not only accurate but also aligned with domain-specific reasoning processes. We hope this encourages further research into interpretable and clinically informed learning frameworks, contributing to the safe and effective integration of AI into healthcare practice.

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

# A  Appendix

## A.1  Data Preparation

---
**Algorithm 2:** Radiology Report → Horn Rule

---
**Input: Input:** $R$:Radiology report
**Output:  Output:** Horn rule $h$
Extract entities $\mathcal{E}$ from $R$ using Stanza i2b2 NER
Initialize empty set of predicates $\mathcal{P}$
**foreach do**
    each entity $e \in \mathcal{E}$
Search within $\pm 10$ tokens around $e$ in $R$
Identify modifiers $m \in \{\text{Negation, Intensity, Location}\}$
Form entity–modifier pair $(e, m)$
Convert $(e, m)$ into predicate $p = \texttt{Predicate}(e, m)$
Add $p$ to $\mathcal{P}$
Initialize impression as clinical conclusion $q$ from predicates $\mathcal{P}$
Construct Horn rule: $h = p_1 \wedge p_2 \wedge \cdots \wedge p_k \;\; \rightarrow \;\; q$
**return** $h$

---

## A.2 LLM-Prompt

> Your evaluation must adhere to the following sub-dimensions of similarity:
>
> 1. **Factual Equivalence:** Do both reports describe the same medical findings?
>
> 2. **Completeness:** Does the generated report capture all key information from the ground truth?
>
> 3. **Absence of Contradiction:** Does the generated report introduce information that conflicts with the ground truth?
>
> **Reports:**
>
> - Ground Truth Report: `{ground_truth_text}`
>
> - Generated Report: `{vlm_output_text}`
>
> **Likert Scale for Similarity:**
>
> - 5: Strongly Agree
>
> - 4: Agree
>
> - 3: Neutral
>
> - 2: Disagree
>
> - 1: Strongly Disagree
>
> **Task:** Provide a JSON object with the evaluation:
>
> ```
> {
>   "similarity_score": <1-5>,
>   "rationale": "<brief justification>"
> }
> ```

### A.2.1 Report-Rule dataset

Following the rule extraction procedure described in Algorithm 2, each radiology report is paired with its corresponding clinical rule and labeled as aligned (label 1), indicating that the report satisfies the logical relationship encoded by the rule. To create negative examples, we generate dispreferred or altered versions of the same report that violate the rule, labeling these as misaligned (label 0). This process results in a dataset of aligned and misaligned report–rule pairs, providing explicit supervision for training the neural verifier to distinguish clinically consistent outputs from inconsistent ones, thereby grounding the model in interpretable clinical logic. Additionally, we test our verifier on a held out test-set and the precision, recall and F1 scores are 0.9863, 0.9730 and 0.9816.

### A.2.2 Preference Data Curation

For preference optimization to be effective, the underlying pre-trained model must be capable of generating at least one output that can be considered a preferred sample. Building on this principle, we adopt a sampling-based strategy: for each model under consideration, we generate multiple candidate reports and then employ a pre-trained verifier to score and rank them. From this ranked list, we select the two highest-scoring candidates, designating the top-scoring report as the *preferred* sample and the second-highest as the *dispreferred* sample.

In terms of dataset preprocessing, we apply this procedure comprehensively across the entire IU-X dataset, extracting structured rules for all samples to support rule-guided supervision. For the MIMIC-CXR-JPG

dataset, due to its large scale, we restrict this process to a randomly selected subset of 10,000 samples while maintaining the same rule-extraction pipeline. This design choice is justified by the fact that both Med-VLMs used in our experiments have been previously exposed to large-scale medical datasets, which facilitates generalization even when subsampling.

For general-purpose pre-trained vision–language models that have not been trained on medical data, we perform an additional supervised fine-tuning (SFT) step on paired image–text data before applying preference optimization. This ensures that the model develops a minimal level of domain alignment.

Apart from that, we use the official test split for both the datasets.

### A.3 Generated Outputs

The evaluation of the candidate report reveals that it includes a single false finding, specifically mild midthoracic dextroscoliosis, which was incorrectly reported. Apart from this, the report does not omit any findings present in the reference, misidentify anatomical locations, or misassess the severity of findings, indicating strong overall accuracy in capturing the reported observations. There are also no instances of mentioning comparisons that are absent in the reference, reflecting careful adherence to the source content. However, the report does omit a comparison detailing changes from a prior study, which could be clinically relevant for tracking patient progression. Overall, while the report demonstrates high fidelity to the reference, and shows the efficacy of our method to capture the linguistic reporting style along with maintaining clinical grounding, by keeping the entities intact. Additionally, we analyze representative negative samples to identify common failure modes of the model. These examples reveal that the model can generate clinically plausible but unsupported findings, including unnecessary specificity and hallucinated abnormalities that are absent from the reference reports. Such errors likely arise from the model relying on learned associations among frequently co-occurring radiological patterns rather than strictly grounding generated statements in case-specific evidence. Although the generated reports remain fluent and coherent, they may violate clinical consistency constraints, highlighting the need for verifier-based supervision to identify and penalize unsupported clinical relations.

> **Zero-Shot**: findings:heart size, mediastinal and hilar contours are normal. lungs are clear. no pleural effusion or acute skeletal abnormality. impression: no radiographic evidence of pneumonia.
> **DPO**: findings: heart size, mediastinal and hilar contours are normal. lungs and pleural surfaces are clear. mild s-shaped scoliosis is noted. impression: no radiographic evidence of pneumonia. impression: No acute findings.
> **SCOPE-RRG**: findings: heart size, mediastinal and hilar contours are normal. lungs and pleural surfaces are clear. scoliosis is demonstrated. impression: no acute cardiopulmonary radiographic abnormality.This is a chest x-ray. The heart is normal size, the mediastinum is normal, and the lungs are clear.

**Generated**: findings: small right apical pneumothorax. multiple right-sided rib fractures are visualized. there is a small amount of pleural fluid tracking along the fissure. the lungs are clear. the cardiomediastinal silhouette is within normal limits. impression: small right apical pneumothorax, multiple right-sided rib fractures, and pleural fluid.

**Ground Truth**: findings:there are low lung volumes with associated bronchovascular crowding and basilar subsegmental atelectasis there is stable prominence of the right cardiac silhouette the mediastinum is unremarkable there is no pleural effusion, pneumothorax, or focal airspace disease there are stable chronic degenerative changes of the thoracic spine impression:no acute cardiopulmonary abnormality

**Generated**: findings: large right pneumothorax with basilar and apical components. there is no evidence of mediastinal shift. opacity in the right lung base is present, likely reflecting a combination of atelectasis and localized contusion. multiple right-sided rib fractures are demonstrated. small right pleural effusion is also demonstrated.

**Ground Truth**: findings:the heart is enlarged the mediastinal contours are stable large hiatal hernia is identified containing stomach and colon questionable small bowel loops are also visualized inthe large hernia there is adjacent compressive atelectasis the lungs are clear impression:no acute disease stable appearance of large hiatal hernia containing stomach and large bowel as well as possible small bowel loops.

**Generated**: findings: there is enlargement of the cardiac silhouette with tortuosity of the aorta. no definite vascular congestion or acute pneumonia. midline sternal wires are intact. scoliosis of the thoracic spine is noted. impression: cardiomegaly without definite acute vascular congestion or pneumonia., **Ground Truth**: findings:stable cardiomegaly sternotomy are intact no pneumothorax or pleural effusion calcific density in the left mid to upper lung represents old granulomatous disease no focal consolidation stable moderate thoracic levoscoliosis and mild thoracolumbar dextroscoliosis impression:stable cardiomegaly with no focal airspace disease stable moderate thoracic levoscoliosis and mild thoracolumbar dextroscoliosis

**Generated**: findings: there is a 3.5 cm round opacity in the left upper lobe. there is no evidence of pneumothorax. the cardiomediastinal silhouette is normal. the imaged upper abdomen is unremarkable. the bones are intact. impression: 3.5 cm round opacity in the left upper lobe. recommend ct chest with contrast for further evaluation.", **Ground Truth**: "findings:lung volumes are low no focal infiltrates heart size normal impression:hypoinflation with no visible active cardiopulmonary disease"

## A.4 Ablation based on noisy rules and fixed lambda

| Models | IUX | | | | | | |
|---|---|---|---|---|---|---|---|
| | BL-1 | BL-2 | BL-3 | BL-4 | RG-L | CBS | RG-F1 |
| Med-Gemma (ZS) | 0.2739±0.11 | 0.1472±0.07 | 0.0719±0.06 | 0.0233±0.05 | 0.2417±0.09 | 0.8769±0.03 | 0.2602±0.13 |
| + DPO | 0.2517±0.09 | 0.1427±0.07 | 0.0748 ± 0.06 | 0.0273±0.04 | 0.2409±0.09 | 0.8477±0.04 | 0.2249±0.12 |
| + RRG-DPO | 0.2189±0.09 | 0.0952±0.07 | 0.0351±0.05 | 0.0122 ±0.03 | 0.1954±0.08 | 0.8507±0.04 | 0.1853±0.12 |
| + MMedPO | 0.2640±0.10 | 0.1358±0.07 | 0.0626±0.06 | 0.0254±0.05 | 0.2363±0.08 | 0.8534±0.04 | 0.2548±0.13 |
| + noisy-rules | 0.2552±0.09 | 0.1525±0.07 | 0.0826±0.06 | 0.0454±0.05 | 0.2509±0.08 | 0.8695±0.04 | 0.2906±13 |
| + fixed-lambda(=0.1) | 0.2652±0.09 | 0.1537±0.07 | 0.0826±0.07 | 0.0404±0.05 | 0.2465±0.09 | 0.8605±0.04 | 0.2783±0.12 |
| + fixed-lambda(=0.5) | 0.2375±0.07 | 0.1334±0.05 | 0.0696±0.05 | 0.0332±0.04 | 0.2209±0.08 | 0.8581±0.04 | 0.2688±0.12 |
| +SCOPE-RRG (Ours) | 0.2936±0.10 | 0.1748±0.08 | 0.0997±0.07 | 0.0539±0.06 | 0.2771±0.10 | 0.8817±0.03 | 0.3100±13 |

Table 6: Comparison of VLMs for radiology report generation across multiple evaluation metrics, including BLEU, and ROUGE score for assessing the lexical accuracy. The table highlights the improvements achieved by our SCOPE-RRG framework baselines with fixed lambda and verifier trained with noisy rules.

In Table 6, we perform an in-depth ablation study to analyze the contribution of different components of the proposed framework. First, to evaluate the robustness of the verifier to rule quality, we construct a noisy preference dataset containing 50% corrupted rules. The noise is introduced by randomly disrupting the

extracted clinical relationships, including jumbling the findings-to-impression ordering and removing selected pathological concepts from the rule structures. We observe a significant degradation in performance across both lexical and clinical metrics when training with noisy rules. This demonstrates that the proposed verifier does not rely solely on entity-level overlap or surface-level matching, but instead leverages the relational semantics captured by the extracted clinical constraints. Second, we investigate the effect of adaptive verifier weighting by comparing it with fixed $\lambda$ settings. We observe that removing the adaptive $\lambda$ strategy leads to a consistent performance decrease across evaluation metrics. This highlights the importance of dynamically balancing the contribution of the verifier-based regularization during optimization, rather than relying on a fixed weighting factor throughout training. These results demonstrate that both high-quality rule supervision and adaptive verifier integration are important components of the proposed framework.

## A.5 Significance Testing

To further validate the robustness of our reported improvements, we conduct paired statistical significance tests using per-sample metric scores obtained on the same evaluation set. Since each generated report is evaluated against the same reference report across all competing methods, paired testing accounts for sample-level variation and provides a more appropriate comparison than independent testing. Specifically, we compare SCOPE-RRG against each baseline for all evaluation metrics and report the corresponding p-values. The significance analysis shows that the improvements achieved by SCOPE-RRG are statistically meaningful across lexical and clinical evaluation metrics, indicating that the performance gains are not attributed to random fluctuations in the test set. This analysis provides additional evidence that the proposed rule-guided preference optimization consistently improves report generation quality beyond baseline approaches.

Table 7: Statistical significance analysis using paired significance tests between SCOPE-RRG and baseline methods. Lower p-values indicate statistically significant improvements.

| Metric | SCOPE-RRG vs MMEDPO | SCOEP-RRG vs RRGDPO | SCOPE-RRG vs DPO | SCOPE-RRG vs ZS |
|---|---|---|---|---|
| BLEU-1 | $7.54 \times 10^{-7}$ | $6.73 \times 10^{-6}$ | $2.33 \times 10^{-4}$ | $3.15 \times 10^{-3}$ |
| BLEU-2 | $5.58 \times 10^{-7}$ | $1.86 \times 10^{-6}$ | $5.45 \times 10^{-5}$ | $2.36 \times 10^{-8}$ |
| BLEU-3 | $1.71 \times 10^{-8}$ | $1.76 \times 10^{-8}$ | $2.69 \times 10^{-3}$ | $1.70 \times 10^{-10}$ |
| BLEU-4 | $2.24 \times 10^{-9}$ | $3.10 \times 10^{-4}$ | $1.52 \times 10^{-8}$ | $4.93 \times 10^{-2}$ |
| ROUGE-L | $8.03 \times 10^{-3}$ | $2.45 \times 10^{-9}$ | $1.24 \times 10^{-7}$ | $1.62 \times 10^{-9}$ |
| RadGraph-F1 | $1.69 \times 10^{-11}$ | $3.02 \times 10^{-5}$ | $1.83 \times 10^{-5}$ | $2.85 \times 10^{-9}$ |
| MedBERT Similarity | $4.82 \times 10^{-4}$ | $4.04 \times 10^{-5}$ | $7.85 \times 10^{-5}$ | $1.63 \times 10^{-3}$ |

We perform one-sided Mann-Whitney U tests using per-sample metric scores to assess the significance of improvements achieved by SCOPE-RRG over baseline methods. The resulting $p$-values are reported in Table 7, showing statistically significant improvements across lexical and clinical metrics.

## A.6 Entity-based vs Clinical Rules as reward

We evaluate our proposed framework against an entity-based reward, RadGraph-F1, to investigate whether entity-relation supervision provides additional benefits beyond entity preservation. In order to perform this we replace the verifier based reward with radgraph. A key difference between RadGraph-F1 and SCOPE-RRG lies in the granularity of the reward signal. RadGraph-F1 primarily measures entity overlap between generated and reference reports, encouraging the preservation of clinical entities but lacking explicit modeling of the relationships among these entities. In contrast, SCOPE-RRG incorporates both entity information and the underlying clinical relations encoded within the clinical rules, enabling the reward signal to capture structured finding–impression dependencies. As shown in Table 8, replacing the proposed verifier-based reward with RadGraph-F1 leads to consistent degradation across evaluation metrics. Notably, despite directly optimizing with RadGraph-F1 as the reward, the RG-F1 score itself decreases substantially compared to SCOPE-RRG. This observation suggests that optimizing entity-level similarity alone is insufficient to preserve the overall clinical coherence and report structure. In contrast, SCOPE-RRG utilizes relational clinical constraints in addition to entity preservation, resulting in improved factual consistency and fluency. These results validate the importance of incorporating structured clinical relationships into the preference optimization framework. We observe this using a qualitative example as follow:

| Rewards | BL-1 | BL-2 | BL-3 | BL-4 | RG-L | CBS | RG-F1 |
|---|---|---|---|---|---|---|---|
| RG-F1 (reward) | 0.2149±0.07 | 0.1177±0.05 | 0.0572±0.04 | 0.0231±0.03 | 0.2028±0.06 | 0.8681±0.04 | 0.2518±0.11 |
| SCOPE-RRG (Ours) | 0.2936±0.10 | 0.1748±0.08 | 0.0997±0.07 | 0.0539±0.06 | 0.2771±0.10 | 0.8817±0.03 | 0.3100±13 |

Table 8: The table highlights the improvements achieved by our SCOPE-RRG framework compared to using RadGraph-F1 as a reward.

1. *Ground Truth*: *findings: normal cardiomediastinal silhouette there is no focal consolidation there are no of a large pleural effusion there is no pneumothorax there is no acute bony abnormality seen impression: there is no radiographic evidence of acute cardiopulmonary disease.*

2. *RadGraph-F1 (reward)*: *findings: cardiomediastinal contours are normal. lungs are grossly clear. small left pleural effusion is noted. no pneumothorax. small left pleural effusion of the chest. the heart size is normal. the mediastinum is normal. the lungs are clear. impression: there is a small left pleural effusion. there is no pneumothorax. the bony structures are unremarkable. the overall appearance is normal.*

3. *SCOPE-RRG*: *findings: there is no evidence of pneumothorax. cardiomediastinal contours are normal. lungs are clear. there is no evidence of pleural effusion. impression: no acute cardiopulmonary process*

This example highlights the limitation of optimizing solely for entity-level similarity. The RadGraph-F1 optimized report preserves several overlapping entities with the reference but introduces an unsupported pathological finding (e.g., small left pleural effusion), leading to a clinically incorrect impression despite achieving strong entity matching. Also, the output does not follow the proper findings impression format due to the missing entity relationship signal. In contrast, the SCOPE-RRG optimized report maintains the correct relationships among findings and impressions, preserving the normal study interpretation without introducing additional abnormalities. This example demonstrates that relational rule-based supervision provides a more clinically meaningful optimization signal by encouraging for entity preservation and relationships. This particular experiment showcases the utility of the rule structure, instead of focused entity matching. The example helps us conclude that in the RadGraph-F1 setting the model is forced to focus on entity matching solely, which disorients the entity relationships, hampering report quality and consistency.

## B   Gradient Analysis of SCOPE-RRG

We analyze the optimization properties of the proposed SCOPE-RRG objective. The training objective consists of a preference optimization term regularized by symbolic verifier feedback:

$$\mathcal{L}_{\text{SCOPE-RRG}}(\theta) = L_{\text{pref}}(\theta) + \lambda L_{\text{rule}}(\theta), \tag{7}$$

where

$$L_{\text{pref}}(\theta) = \mathbb{E}_{(x,y_w,y_l)\sim\mathcal{D}_P}\left[-\log\sigma\left(\beta\Delta_\theta\right)\right], \tag{8}$$

with

$$\Delta_\theta = \log\pi_\theta(y_w|x) - \log\pi_\theta(y_l|x), \tag{9}$$

and the symbolic verifier regularization term is

$$L_{\text{rule}}(\theta) = \mathbb{E}_{(y_e,r)\sim\mathcal{D}_P}\left[1 - V_\theta(y_e,r)\right]. \tag{10}$$

Here $V_\theta(y_e, r) \in [0, 1]$ denotes the neural verifier's rule-consistency score, and $\lambda \geq 0$ controls the strength of symbolic feedback.

**Notation.** For compactness, let

$$z = \beta \Delta_\theta \tag{11}$$

so that

$$L_{\text{pref}} = \mathbb{E}[-\log \sigma(z)]. \tag{12}$$

**Lemma 1** (Gradient of the preference loss)**.** *The derivative of* $-\log \sigma(z)$ *with respect to* $z$ *is*

$$\frac{\partial}{\partial z}[-\log \sigma(z)] = -(1 - \sigma(z)). \tag{13}$$

*Proof.* Using

$$\sigma'(z) = \sigma(z)(1 - \sigma(z)), \tag{14}$$

we obtain

$$\frac{\partial}{\partial z}[-\log \sigma(z)] = -\frac{\sigma'(z)}{\sigma(z)} = -\frac{\sigma(z)(1 - \sigma(z))}{\sigma(z)} = -(1 - \sigma(z)). \tag{15}$$

□ □

**Proposition 1** (Full generator gradient)**.** *The gradient of the SCOPE-RRG objective with respect to model parameters* $\theta$ *is*

$$\nabla_\theta \mathcal{L}_{SCOPE\text{-}RRG} = -(1 - \sigma(z))\beta\Big(\nabla_\theta \log \pi_\theta(y_w|x) - \nabla_\theta \log \pi_\theta(y_l|x)\Big) + \lambda \nabla_\theta L_{rule}. \tag{16}$$

*Proof.* Differentiating Eq. (7) gives

$$\nabla_\theta \mathcal{L} = \nabla_\theta L_{\text{pref}} + \lambda \nabla_\theta L_{\text{rule}}. \tag{17}$$

Applying the chain rule to the preference term:

$$\nabla_\theta L_{\text{pref}} = -(1 - \sigma(z))\beta\Big(\nabla_\theta \log \pi_\theta(y_w|x) - \nabla_\theta \log \pi_\theta(y_l|x)\Big). \tag{18}$$

Substituting into the joint gradient yields Eq. (16).

□ □

*Remark* 1 (Gradient decomposition). The gradient in Proposition 1 naturally decomposes into two complementary terms.

- **Preference Gradient**

$$-(1 - \sigma(z))\beta\Big(\nabla_\theta \log \pi_\theta(y_w|x) - \nabla_\theta \log \pi_\theta(y_l|x)\Big) \tag{19}$$

  is identical in form to SimPO and drives preference alignment by increasing the likelihood of preferred reports while suppressing rejected ones.

- **Symbolic Feedback Gradient**

$$\lambda \nabla_\theta L_{\text{rule}} \tag{20}$$

acts as a corrective signal encouraging generations that satisfy clinical rule structure.

Rather than modifying the preference margin directly, symbolic feedback perturbs the optimization trajectory through additive regularization, steering learning away from shortcut lexical solutions and toward clinically grounded generations.

*Remark* 2 (Role of the adaptive weighting coefficient). The parameter $\lambda$ governs the trade-off between linguistic preference alignment and symbolic consistency.

Large $\lambda$ increases the influence of rule feedback and prioritizes clinical correctness, while smaller $\lambda$ recovers behavior closer to standard preference optimization.

Dynamic updates of $\lambda$ allow the symbolic signal to strengthen when rule violations increase and relax when consistency improves, preventing either objective from dominating optimization.

## B.1 Experimental Setup

### B.1.1 Neural Verifier Training

Table 9: Experimental setup for Neural Verifier Training.

| Component | Configuration |
|---|---|
| **Base Model** | RoBERTa-base ("Riiid/kda-roberta-base-race") |
| **Classifier Head** | 2-layer MLP + ReLU + Dropout, binary output |
| **LoRA Config** | $r = 8$, $\alpha = 16$, target modules = {query, key}, dropout=0.05 |
| **Tokenizer** | RobertaTokenizer, with padding |
| **Dataset** | RuleVerifierDataset (MIMIC, JSON with rule–report pairs) |
| **Task** | Binary classification (rule consistency) |
| **Batch Size** | 128 |
| **Optimizer** | AdamW, learning rate $1 \times 10^{-4}$, weight decay=0.01 |
| **Scheduler** | Cosine schedule, warmup ratio 5% |
| **Loss Function** | Cross-Entropy Loss |
| **Epochs** | 120 |
| **Metrics** | Precision, Recall, F1-score, Accuracy (micro-averaged) |
| **Device** | NVIDIA GPU (CUDA) |

### B.1.2 DPO Training

Table 10: Experimental setup for DPO fine-tuning.

| Component | Configuration |
|---|---|
| **Backbone Model** | Med-Gemma (4B, multimodal, Image-Text-to-Text). |
| **LoRA Configuration** | Rank $r = 8$, $\alpha = 16$, dropout=0.1, applied to `q_proj` and `v_proj`. |
| **Dataset** | Indiana University Chest X-ray (paired with rule-annotated reports). JSON annotation with chosen vs. rejected samples. |
| **Image Preprocessing** | Resize 224×224, normalization mean=[0.5], std=[0.5], RGB conversion. |
| **Tokenizer / Processor** | HuggingFace AutoProcessor (Google Med-Gemma). |
| **Loss Function** | Direct Preference Optimization (DPO) loss with $\beta = 0.1$. |
| **Optimizer** | Adam, learning rate $= 5 \times 10^{-4}$. |
| **Scheduler** | Linear scheduler with 5% warm-up steps. |
| **Batching** | Batch size $= 2$, gradient accumulation $= 8$ (effective batch $= 16$). |
| **Precision** | Mixed precision training with bfloat16 + gradient scaling. |
| **Epochs** | 20 epochs. Model checkpoint saved at epoch 20. |

### B.1.3 SCOPE-RRG

Table 11: Experimental setup for SCOPE-RRG training.

| Component | Configuration |
|---|---|
| **Base Model** | Med-Gemma (conditional generation) with LoRA (rank $= 8$, $\alpha = 16$, dropout $= 0.1$) applied to `q_proj` and `v_proj`. |
| **Verifier Model** | RoBERTa-base (Riiid/kda-roberta-base-race) with custom binary classifier head and LoRA (rank $= 8$, dropout $= 0.05$) on query/key projections. |
| **Dataset** | Indiana University Chest X-ray dataset; normalized PNGs with paired reports and predicate rules. |
| **Image Preprocessing** | Resize to 224×224, normalization to [-1, 1]. |
| **Optimization (Generator)** | Adam optimizer, learning rate $= 5e-4$, linear scheduler with 5% warm-up. |
| **Optimization (Verifier Constraint)** | Adam optimizer on Lagrange multiplier, learning rate $= 4e-2$. |
| **Batching** | Batch size $= 2$, gradient accumulation $= 8$ (effective batch $= 8$). |
| **Training Duration** | 15 epochs, checkpoint every 2 epochs. |
| **Precision** | Mixed precision with bfloat16 and gradient scaling. |

### B.2 Loss Description and Decoding Setting

During training, the verifier loss is computed from reports sampled from the current policy model. For each input image and prompt, the model generates a candidate report using stochastic decoding with top-$p = 0.9$, temperature $= 0.6$, and a maximum generation length of 100 tokens. The generated sequence is decoded into text and paired with the corresponding extracted clinical rule to form a $(rule, report)$ pair, which is then provided as input to the verifier. The verifier produces a consistency score that is used to compute the verifier loss.

Since the generated report is a discrete sequence, the sampling operation is non-differentiable and gradients are not propagated through the generation step. Instead, the verifier loss provides an auxiliary supervision signal that is jointly optimized with the DPO objective. The overall optimization objective is defined as:

$$\mathcal{L} = \mathcal{L}_{\text{DPO}} + \lambda \mathcal{L}_{\text{verifier}}, \tag{21}$$

where $\lambda$ controls the contribution of the verifier supervision. To avoid negative weighting and allow adaptive balancing during optimization, we parameterize $\lambda$ using a softplus transformation:

$$\lambda = \text{softplus}(\lambda_{\text{mul}}). \tag{22}$$

The combined objective is optimized through backpropagation, updating the parameters of the vision-language model using the DPO and verifier-based supervision signals.

