# OpenReview forum: "SCOPE-RRG: Symbolic Constraint Preference Optimization for Radiology Report Generation"
_TMLR — Under review for TMLR_

### Review · Reviewer_GqcM · 2026-05-22

**Summary Of Contributions:**

This paper proposes SCOPE-RRG, a symbolic-constraint-guided preference optimization framework for radiology report generation. Motivated by the observation that prior methods often treat report generation as a next-token prediction task and do not explicitly model the reasoning link between findings and impressions, the authors extract natural-language Horn rules from radiology reports and use them as clinical constraints. A neural verifier is trained to assess whether a generated report is aligned with the corresponding rule, and its feedback is used both for preference data construction and joint optimization. The goal is to guide VLMs toward generating reports that are linguistically fluent while remaining clinically coherent and faithful to structured medical knowledge.

**Audience:**

Yes

**Audience Explanation:**

The paper addresses an important and timely problem: improving the factual and clinical consistency of radiology report generation. Given the growing interest in reliable and trustworthy VLMs, I believe at least a subset of the TMLR audience would find the findings useful.

**Claims And Evidence:**

Yes

**Claims Explanation:**

1. The submission reports improvements on both IU-Xray and MIMIC-CXR, using lexical metrics such as BLEU/ROUGE and clinical/semantic metrics such as ClinicalBERTScore, RadGraph-F1, and GREEN.

2. The experiments include multiple base VLMs, including Med-Gemma, Med-Flamingo, and LLaVA-Rad. This makes the evidence more convincing.

3. The authors train the verifier on one dataset and evaluate it on the other dataset, reporting that verifier-guided performance transfers across MIMIC and IU-Xray.

**Requested Changes:**

1. The proposed method is motivated as a way to reduce clinical hallucination and improve clinical correctness by incorporating symbolic clinical constraints. However, the reported improvements appear more pronounced on standard NLG metrics such as BLEU and ROUGE than on clinically oriented metrics such as RadGraph-F1 and GREEN. The authors should discuss this discrepancy more explicitly.

2. Some of the paper’s broader claims, such as reduced hallucination, improved clinical reasoning, and suitability for trustworthy real-world deployment, are currently supported by quantitative evaluation.

3. The optimization objective appears closely related to SimPO. The main difference seems to be that the preference pairs and the additional regularization signal are derived from a neural verifier trained on symbolic clinical rules. While this is a potentially useful adaptation to radiology report generation, the paper should more clearly distinguish the algorithmic contribution from simply replacing the preference judge/reward model in a SimPO-like framework with a domain-specific verifier.

4. The paper repeatedly describes the proposed method as producing “clinically grounded” medical VLMs, but the notion of grounding is not clearly defined. The neural verifier checks whether a generated report is aligned with an extracted symbolic rule, but it does not directly verify whether the generated clinical entities are supported by the underlying X-ray image.

---

> ### Author Response · Authors · 2026-06-09
>
> ### Why Are the Improvements More Pronounced on BLEU and ROUGE Than on Clinical Metrics Such as RadGraph-F1 and GREEN? (Q1)
> - We thank the reviewer for pointing out this important nuance. Our proposed framework achieves noticeable improvement across semantic metric such as RadGraph-F1, while the improvement in GREEN is not equivalently pronounced, majorly due to the different output dimensionality captured by these two metrics. RadGraph-F1 primarily measures the correctness of clinical entities and relations extracted from the generated report. Since our method explicitly incorporates symbolic clinical constraints that guide the generation of clinically structured outputs that are linguistically and coherently grounded, the outputs naturally align with the evaluation criteria of RadGraph-F1. Conversely, the proposed structural constraint encourages better entity-relation consistency, directly reflected by RadGraph-F1 rather than GREEN.
> GREEN metric performs a diverse evaluation across six dimensions as follows:
>      - false reporting of findings,
>      - omission of clinically relevant findings,
>      - incorrect anatomical localization,
>      - incorrect severity assessment,
>      - unsupported comparative statements, and
>      - omission of temporal progression comparisons.
> Our current symbolic constraint are primarily designed as per standard reporting guidelines. As a result, improvements in structural clinical correctness do not directly translate into higher GREEN scores. For example, a report may correctly identify a pathology and therefore improve RadGraph-F1, while still being penalized by GREEN for subtle issues such as omission of temporal progression details. Furthermore, several GREEN error categories depend on longitudinal or comparative reasoning which is not focused by the proposed framework. GREEN provides a holistic evaluation criteria, where improvements in one error category may be offset by residual errors in others.
> We will revise the manuscript to explicitly clarify that the proposed symbolic constraint framework is particularly effective at improving structured and relational consistency, which is more directly reflected in RadGraph-F1, while GREEN evaluates a broader spectrum of reporting errors.
>
> ### How does the proposed optimization differ from SimPO beyond replacing the preference judge with a neural verifier trained on symbolic clinical rules? Is the contribution algorithmic, or primarily an application-specific adaptation of a SimPO-like framework
>
> - We thank the reviewer for this insightful comment. As rightly pointed out, SCOPE-RRG shares conceptual similarities with SimPO, as our framework builds upon preference optimization principles and leverages the notion of target reward margins. However, we would like to clarify that our contribution extends beyond simply replacing the reward margin with a domain-specific verifier and instead constitutes a task-specific extension of the preference optimization framework.
>
> Our framework integrates symbolic clinical reasoning directly into preference optimization. While standard SimPO relies on a fixed scalar target reward margin, SCOPE-RRG defines the margin adaptively using a frozen neural verifier trained on symbolic clinical rules. In a standard SimPO objective, $\gamma$ is a fixed constant margin applied uniformly across all preference pairs.
> In contrast, SCOPE-RRG uses a clinically conditioned adaptive margin:
> $\gamma_{r}=Verifier(y_e,r)\in[0,1]$
> Further, rule consistency is explicitly enforced through the regularization term:
> $L_{sym}(\theta)= \mathbb{E}_{(x,y_e)\sim\mathcal{D}_P}
> \left[
> 1 - Verifier(y_e,r)
> \right].$, as per Eq. (4). which directly penalizes reports that are poorly aligned with symbolic clinical constraints.
> Intuitively, SCOPE-RRG dynamically adjusts the required margin based on the report’s current clinical consistency, unlike a fixed margin. This means that reports with stronger rule alignment are required to achieve larger preference separation, while clinically inconsistent generations incur higher regularization penalties. This creates progressively stricter clinically grounded optimization pressure.
> The verifier therefore serves not merely as a substitute reward model, but as a supervisory mechanism operating over clinically interpretable rules and relations derived using standard reporting guidelines.
> From this perspective, SCOPE-RRG is best viewed as a **task-specific extension through rule-guided adaptive margin regularisation**. The novelty lies not simply in replacing the reward model, but in introducing clinically grounded dynamic preference boundaries and explicit rule-consistency regularization for radiology report generation.
>
> We will revise the manuscript to more clearly position SCOPE-RRG relative to SimPO and redo the mathematical equations for better explanations and clear distinction.

---

> > ### Author Response · Authors · 2026-06-09
> >
> > ### Clinical Grounding Is Not Explicitly Established (Q4)
> > - We thank the reviewer for this important observation. Our use of this term refers to structural and semantic grounding rather than explicit visual grounding. We define a reduced symbolic structure of a radiology report as per standard reporting guidelines: $findings\implies impression$.
> > Specifically, the proposed verifier does not directly validate whether each generated clinical entity is visually supported by the underlying X-ray image; instead, it evaluates whether the generated report is consistent with symbolic clinical rules derived from radiology entities, and their logical relationships. Since report generation remains conditioned on image representations through the underlying Med-VLM, the verifier acts as an clinically structured supervision. This supervision enforces semantic consistency and discourages clinically implausible generations. While training this clinically structured supervision forces a consistent textual output which indirectly forces the vision encoder to learn representations aligned to the symbolic structure.
> >
> > We will revise the manuscript to explicitly distinguish clinical semantic grounding from visual evidence grounding, clarify this limitation, and note that integrating region-level visual verification would be an important direction for future work.

---

### Review · Reviewer_EPf1 · 2026-05-26

**Summary Of Contributions:**

This paper proposes SCOPE-RRG, a preference optimization framework for radiology report generation that incorporates symbolic constraints derived from the structural relationship between findings and impressions. The authors extract entity–modifier predicates from each report via Stanza i2b2 NER and form natural-language Horn rules of the form findings to impression. A RoBERTa-based neural verifier is trained to predict whether a (report, rule) pair is aligned, and is then used in two roles: (i) to curate preference pairs by ranking sampled VLM outputs, and (ii) to provide a symbolic regularization signal jointly optimized with a preference loss under an adaptive Lagrangian weight. Experiments on MIMIC-CXR-JPG and IU-Xray across three Med-VLM backbones (Med-Gemma, Med-Flamingo, LLaVA-Rad) report gains over zero-shot, DPO, RRG-DPO, and MMedPO baselines on both lexical and semantic metrics, accompanied by verifier cross-institution and sensitivity ablations.

**Audience:**

Yes

**Audience Explanation:**

Medical report generation is a hot topic in both medical Ai and vision-language communities.

**Broader Impact Concerns:**

The submission includes a Broader Impact Statement, but it overstates the safety properties of the proposed system in ways that warrant revision rather than additional content per se.

First, the statement frames SCOPE-RRG as reducing hallucinations and improving "factual quality" and "clinical grounding." Given that the verifier operates on text only and that the supervision signal is derived from entities extracted from the ground-truth reports themselves, the framework does not actually verify alignment with the underlying image. A report that is structurally consistent with the derived rule but describes findings absent from the X-ray would still receive a high symbolic-loss reward. Presenting such a system as enhancing trustworthiness for "high-stakes medical settings" risks overstating its safety guarantees to downstream readers and potential adopters. The statement should explicitly scope the claim to textual/structural consistency and note that visual faithfulness is not enforced by the current design.

Second, the statement suggests the system could serve as an assistive tool for radiologists. Given the regressions and small-magnitude improvements on clinically meaningful metrics (RadGraph-F1, GREEN, ClinicalBERTScore) in several settings, and the absence of any prospective or reader study, the deployment framing is premature. The authors should temper this language and explicitly state that the current evaluation does not support clinical deployment claims.

Finally, there is no discussion of patient-population coverage or failure-mode asymmetry.

**Claims And Evidence:**

No

**Claims Explanation:**

Several of the paper's central claims are not adequately supported by the reported evidence.

First, the claim that SCOPE-RRG provides symbolic / rule-based / clinically grounded supervision is overstated. The "Horn rules" are extracted from the same report they later supervise, so the verifier effectively learns an entity–modifier consistency check between a report and its own derived structure rather than reasoning over generalizable clinical rules. The framing as symbolic reasoning is not justified by the actual mechanism.

Second, the claim of clinical grounding is undermined by the fact that the verifier is text-only: it takes (rule, report) as input and never observes the image. A generation can therefore receive a high symbolic-loss reward while being visually ungrounded, which directly contradicts the paper's motivation.

Third, the empirical claim of consistent improvements across lexical and semantic metrics is not borne out by the tables. On MIMIC-CXR-JPG, RadGraph-F1 decreases for Med-Gemma (26.02 → 18.18), and CBS decreases for LLaVA-Rad (90.01 → 89.88); several BLEU-3/4 entries for Med-Flamingo fall below zero-shot. GREEN differences (≤0.05) lie within plausible noise. No standard deviations, multiple runs, or significance tests are reported.

Fourth, the cross-institution analysis (Table 3) shows OOD verifiers outperforming in-domain verifiers (e.g., IUX-trained verifier scoring BL-1 = 30.77 on MIMIC vs. 22.90 in-domain), which is counterintuitive and left unexplained; the table also reuses numbers from the main results without clarifying the experimental setup.

Fifth, the verifier-sensitivity ablation (Table 4) begins from a 10%-accuracy verifier, worse than random for a binary task, making it an inverted, not weak, classifier and rendering the "increasing verifier strength improves results" claim unsupported in the regime that actually matters (50–98%).

Finally, key design choices are underspecified: the construction of misaligned (report, rule) pairs for verifier training is not described (Appendix A.2.1 only says reports are "altered"), and no SimPO baseline is reported despite SimPO being repeatedly invoked as motivation, leaving the contribution of the symbolic term relative to a reference-free preference baseline unclear.

**Requested Changes:**

1. Reframe the "symbolic / Horn rule" claim. The current mechanism is an entity–modifier consistency check between a report and its own derived structure, not symbolic reasoning over generalizable clinical rules. Either (a) re-scope the contribution as structured entity-preservation supervision and remove the symbolic-reasoning framing and Clark et al. (2021) framing, or (b) demonstrate that the rules generalize across reports.

2. Address the tautology between rule extraction and supervision. Because rules are derived from the ground-truth reports and the verifier is trained on these same derivations, the gains may largely reflect entity overlap with the GT rather than clinical correctness. Please add (a) an ablation that replaces the verifier signal with a simple entity-preservation loss (e.g., CheXpert/RadGraph entity-weighted CE) and (b) evaluation under a clinical metric whose extractor is not derived from the same NER pipeline used to build the rules.

3. Make the verifier image-aware, or weaken the "clinical grounding" claim. The current text-only verifier cannot distinguish reports that are structurally consistent with the rule but visually wrong. Either add a multimodal verifier variant (image + rule + report) and show the symbolic term still helps, or restate the contribution as textual consistency rather than visual/clinical grounding.

4. Specify and justify the construction of misaligned (rule, report) pairs. Appendix A.2.1 only states that reports are "altered." Please describe the procedure precisely, report verifier accuracy on each negative type, and demonstrate that the negatives are not trivially separable.

5. Report variance and significance.

6. Redesign the verifier sensitivity ablation (Table 4). A 10%-accuracy binary classifier is worse than random and effectively inverted; it does not represent a weak verifier.

---

> ### Author Response · Authors · 2026-06-18
>
> ### Clarification of Symbolic Rule Representation and Rule Generalization
> - We respectfully disagree with the reviewer’s characterization that our method is limited to an entity–modifier consistency check. Our approach is directly motivated by the framework introduced by Clark et.al. in Transformers as Soft Reasoners over Language, which demonstrates that neural language models can learn to reason over symbolic structures represented in natural language without requiring an explicit symbolic inference engine. Following this perspective, we encode radiological knowledge as Horn-style rules derived from clinical reports and train a neural verifier to evaluate whether generated reports satisfy these structured constraints within a differentiable optimization framework. Importantly, the role of the rules in our framework is not limited to preserving individual entities or modifiers. The extracted rules capture relational dependencies between clinical observations and their corresponding interpretations, such as how specific findings are associated with diagnostic impressions. The verifier learns to assess the consistency of generated reports with these structured relationships, providing a learning signal that encourages clinically coherent reasoning patterns.
> Consistent with the soft reasoning paradigm proposed by Clark et al., the goal is not to perform hard symbolic deduction, but rather to integrate symbolic representations as an intermediate form of structured knowledge that guides neural generation. The verifier therefore acts as a neural soft reasoner over clinical constraints, where the symbolic rules provide interpretable relational structure and the neural model learns to evaluate and incorporate these constraints during report generation. This distinguishes our approach from simple entity-preservation supervision, as the supervision signal is derived from structured relationships among clinical concepts rather than isolated entity overlap.
>
> ### Validation of Rule-Based Supervision Beyond Entity Preservation
> - We respectfully disagree that the observed gains can be explained solely by entity overlap with the ground-truth reports. While the rules are extracted from reference reports, the verifier is not trained to match entities directly; rather, it learns to assess the satisfaction of structured clinical constraints expressed as entity–modifier relationships. This directly represents the soft reasoning ability of transformers as per Clark et.al. Consequently, the supervision signal captures consistency at the level of clinical relations rather than simple entity preservation.
> We also note that part (b) of the reviewer's suggestion is already addressed in our evaluation. The clinical metrics used in the paper, including GREEN and RadGraph-F1, rely on extractors that are independent of the Stanza-based pipeline used for rule extraction. Therefore, the reported improvements are measured using evaluation frameworks that are not derived from the same extraction process used to construct the symbolic rules, reducing the possibility of evaluation bias. Additionally, our cross-dataset verifier experiment shows that the training strategy is focused on learning the rule structure rather than becoming dependent on a particular dataset structure or origin.
> Moreover, our evaluation already includes metrics such as GREEN and RadGraph-F1, which are independent of the verifier and are designed to assess clinical correctness beyond lexical overlap. The improvements observed on these metrics suggest that the benefits of the proposed method are not limited to reproducing entities present but retain the entity relationships. We will clarify this distinction in the manuscript and further discuss why the verifier objective provides a richer signal than standard entity-matching losses. While comparisons against alternative entity-preservation objectives are interesting future directions, we believe the current results already demonstrate improvements under clinical evaluation frameworks that are independent of the rule-based supervision used during training.
>
> Additionally, we will add the results using RadGraph-F1 as the reward. However, the experiment did not complete within the stipulated time therefore we request as extension to be able to add the suggested experiment.

---

> > ### Author Response · Authors · 2026-06-18
> >
> > ### Scope of Clinical Grounding
> > - We agree that incorporating image information into the verifier could provide a stronger form of grounding and is an interesting direction for future work. However, we would like to clarify that the primary contribution of this paper is not the design of a multimodal verifier. Rather, our focus is on investigating whether symbolic clinical constraints can be incorporated into preference optimization through a neural verifier inspired by the soft reasoning paradigm.
> > In our framework, image grounding is already provided by the underlying vision-language model, while the verifier serves a different purpose: evaluating the consistency of generated reports with structured clinical rules. Consequently, the verifier is intentionally text-based and is not intended to function as an independent image–report alignment model. We therefore do not claim that the verifier itself provides visual grounding or can detect all cases where a report is consistent with the rules yet inconsistent with the image.
> > Also, we like to bring to attention that the current formulation is sufficient to demonstrate the central hypothesis of the paper—that symbolic clinical constraints can provide a useful supervisory signal during preference optimization. Extending the verifier to jointly reason over images, reports, and rules is a natural and promising direction for future work.
> >
> > ### Misaligned Rule–Report Pairs and Verifier Robustness Analysis
> > - We thank the reviewer for highlighting the importance of clarifying both the rule construction and misaligned pair generation procedures. The extracted rules are not arbitrary templates; they are derived from clinically meaningful concept–modifier relationships identified from radiology reports following the rule extraction procedure described in Appendix A.1. These rules encode structured relationships between observed findings and corresponding impressions, and were designed to capture clinically relevant dependencies rather than simple entity co-occurrence. The extracted rule representations were additionally verified for clinical validity to ensure that the encoded relationships reflect meaningful radiological associations. For verifier training, aligned pairs are created by associating each report with its corresponding extracted rule, where the report satisfies the logical relationship encoded by the rule. Misaligned pairs are generated through controlled alterations of the original reports that violate the associated rule constraints, including disrupting finding–impression relationships, removing relevant pathological concepts, or introducing inconsistent clinical statements. These modifications preserve the overall report structure while creating targeted rule violations, preventing the verifier from relying solely on superficial lexical or formatting cues. We evaluate the verifier on a held-out test set and obtain precision, recall, and F1 scores of 0.9863, 0.9730, and 0.9816, respectively. These results demonstrate that the verifier can effectively distinguish rule-consistent and rule-violating reports. We will further clarify the rule extraction procedure and provide additional details on the negative sample construction and verifier evaluation in the revised manuscript.
> >
> > ### Verifier Sensitivity Analysis and Robustness Evaluation
> > - We thank the reviewer for this suggestion. In the revised manuscript, we report standard deviations across all baselines (Appendix A.4) to better quantify the robustness of the observed improvements, we get a p-value of 0.01 across all baselines which shows that our method is consistent across samples and unlikely to be due to random variations.
> > Regarding the verifier sensitivity analysis, the purpose of the experiment was to illustrate the degradation trend as verifier quality decreases. To address this concern, we will redesign the ablation to focus on more meaningful verifier accuracies and provide a clearer analysis of how generation performance varies with verifier quality. We thank the reviewer for pointing out this limitation and believe the revised experiment will provide a more realistic assessment of verifier sensitivity. However, we were unable to finish the suggested experiment within stipulated time. Therefore, we request for an extension, to provide with the remaining empirical results.

---

> > > ### Author Response · Authors · 2026-06-25
> > >
> > > ## RadGraph-F1 vs Proposed Verifier as reward
> > > - As per the reviewer suggestion we performed the experiment with RadGraph-F1 as reward the results are shown in the table below:
> > > | Rewards | BL-1 | BL-2 | BL-3 | BL-4 | RG-L | CBS | RG-F1 |
> > > |---|---|---|---|---|---|---|---|
> > > | RG-F1 (reward) | 0.2149 ± 0.07 | 0.1177 ± 0.05 | 0.0572 ± 0.04 | 0.0231 ± 0.03 | 0.2028 ± 0.06 | 0.8681 ± 0.04 | 0.2518 ± 0.11 |
> > > | SCOPE-RRG (Ours) | 0.2936 ± 0.10| 0.1748 ± 0.08 | 0.0997 ± 0.07 |0.0539 ± 0.06 | 0.2771 ± 0.10 | 0.8817 ± 0.03 | 0.3100 ± 0.13 |
> > >
> > > The results clearly show a degradation in performance across lexical and semantic metrics. Importantly, we observer a sharp decrease in the RG-F1 metric inspite of using the same as reward, indicating that entity-level optimization alone is insufficient to preserve clinically coherent reports. This highlights the importance of incorporating both clinical entities and the relational dependencies among them, captured by the rule structures $findings \implies impression$. In contrast, SCOPE-RRG leverages structured clinical rules that jointly encode entities and their relationships, providing a richer supervision signal for maintaining clinically consistent findings and impressions. Therefore, SCOPE-RRG differs fundamentally from an entity-preservation objective by explicitly modeling and optimizing relational clinical consistency. We have added the experiment in Appendix A.6 of the revised manuscript.
> > >
> > >
> > >
> > > ## Verifier Sensitivity
> > > - The original sensitivity analysis was intended to study the impact of verifier quality. We have revised with different verifier accuracies at 50%, 70%, 90%. The results show a gradual increase in metrics with the increase of the verifier, which depicts a direct effect of the verifier supervision signal on the final output. The revised results, presented in Section 5.2.2, demonstrate a consistent improvement in performance as verifier quality increases across evaluation metrics. This confirms that the effectiveness of SCOPE-RRG is directly influenced by the quality of the verifier supervision signal and validates the role of the verifier in guiding clinically consistent report generation. The table presents the above discussed results:
> > > | Verifier | BL-1 | BL-2 | BL-3 | BL-4 | RG-L | RG-F1 |
> > > |---|---|---|---|---|---|---|
> > > | SCOPE-RRG, Ver 50\% | 23.84 | 13.62 | 7.32 | 3.69 | 22.65 | 27.80 |
> > > | SCOPE-RRG, Ver 70\% | 24.43 | 13.70 | 6.83 | 3.21 | 22.96 | 26.79 |
> > > | SCOPE-RRG, Ver 90\% | 26.01 | 15.13 | 8.23 | 4.24 | 24.85 | 29.41 |
> > > | SCOPE-RRG, Ver 98\% (Reported) | 29.36 | 17.48 | 9.97 | 5.39 | 27.71 | 31.00 |
> > >
> > > ## Significance Testing
> > > - To further assess the robustness of our reported improvements, we perform paired statistical significance tests using per-sample metric scores obtained on the same evaluation set. We compare SCOPE-RRG against each baseline across all evaluation metrics and report the corresponding $p$-values. The results demonstrate statistically significant improvements across both lexical and clinical evaluation metrics, indicating that the observed gains are unlikely to be due to random variation in the test set. This analysis further supports the effectiveness of the proposed rule-guided preference optimization framework. We report the values in the revised manuscript in Appendix A.5.

---

> > > > ### Author Response · Authors · 2026-06-25
> > > >
> > > > ## Broader Impact Statement
> > > > - We thank the reviewer for raising this important point regarding the scope of our claims. Regarding potential clinical deployment implications, our intention is to position SCOPE-RRG as a step toward developing more reliable radiology report generation systems, rather than as a replacement for radiologists or a clinically deployable solution. We acknowledge that prospective reader studies, extensive clinical validation, and real-world evaluation are necessary before such systems can be considered for clinical use. The improvements observed across complementary lexical and clinical metrics, along with human and LLM-based evaluations, demonstrate the potential of rule-guided preference optimization while not implying immediate clinical adoption.
> > > >
> > > > We would also like to clarify that our broader impact statement discusses the long-term potential of such systems as assistive tools for radiologists, rather than claiming that SCOPE-RRG is deployment-ready. This positioning is consistent with Reviewer BSu6's assessment of the work as a plausible contribution toward the direction of more reliable medical AI systems. We have framed our work as a step in this direction and acknowledge that further validation is required before translation into clinical practice.

---

### Review · Reviewer_BSu6 · 2026-06-04

**Summary Of Contributions:**

The paper proposes SCOPE-RRG, a preference-optimization framework for radiology report generation. Its central idea is to treat the relation between findings and impressions as a clinical constraint, rather than only as text to imitate. The method extracts entity-modifier predicates from reports, forms natural-language Horn-style rules, trains a neural verifier to judge rule consistency, and uses this verifier both to construct preference pairs and to regularize VLM fine-tuning.

The main strength is the attempt to inject an interpretable clinical signal into preference optimization, which is well motivated for radiology report generation. The experiments across MIMIC-CXR-JPG, IU-Xray, and several VLM backbones suggest potential benefits. The main weakness is that the evidence for the rule extraction and verifier is not yet strong enough: rule quality, verifier calibration, statistical robustness, and clinical validation are all under-specified, and the empirical gains are mixed in some settings.

**Audience:**

Yes

**Audience Explanation:**

The paper addresses an important problem: preference optimization can improve fluency while still missing clinically meaningful details. The idea of using symbolic clinical constraints as an interpretable verifier signal is relevant to readers interested in medical VLMs, preference optimization, neuro-symbolic learning, and trustworthy generation. Even though the current evidence needs tightening, the direction is timely and likely to be useful for future work on grounded medical text generation.

**Broader Impact Concerns:**

The paper already includes a broader impact statement, but it is somewhat optimistic. Since the work targets radiology reporting, the authors should explicitly state that the system is intended only as an assistive tool and not for autonomous diagnosis. They should also discuss residual hallucinations, verifier and rule-extraction failures, domain shift across hospitals, dataset bias, and the need for clinician oversight, calibration, and monitoring before clinical use. I do not see additional ethical concerns beyond these medical safety and deployment risks, but these risks should be addressed more directly.

**Claims And Evidence:**

No

**Claims Explanation:**

The main idea is plausible, and the paper provides a useful set of experiments, but several claims are stronger than the evidence. The reported gains are not uniform across all models, datasets, and metrics, so claims of consistent improvement should be softened. More importantly, the method relies heavily on extracted symbolic rules and a neural verifier, but the paper does not sufficiently validate the quality, calibration, or failure modes of either component. The human evaluation is helpful but small, and the paper does not provide enough statistical evidence such as confidence intervals, multiple seeds, or significance tests. Thus, the current evidence supports the direction, but not the stronger claims about reliable or clinically trustworthy report generation.

**Requested Changes:**

1. Critical: The authors should better validate the rule extraction and verifier. This should include a held-out evaluation of rule quality, clinician audit of extracted rules, verifier precision/recall/F1 and calibration, and an analysis of common false positives and false negatives.

2. Critical: The ablations should separate the effects of rule-guided preference data curation and symbolic verifier regularization. A fair comparison should use the same sampled preference pairs with and without the verifier loss, and should also test random/noisy rules and fixed vs adaptive λ.

3. Critical: The optimization details need to be clarified. In particular, the paper should explain how the verifier-based loss is optimized through discrete generated text, and should provide the number of sampled candidates, decoding settings, prompts, split details, random seeds, and MIMIC subset construction.

4. Critical: The clinical evaluation should be strengthened. A larger blinded evaluation with multiple radiologists, baseline comparisons, and clinically meaningful error categories would make the claims much more convincing.

5. Would strengthen: The authors should add confidence intervals or multiple-seed results, discuss the mixed metrics more carefully, and avoid saying the gains are “consistent” where the tables show exceptions.

6. Would strengthen: The paper should discuss the failure cases more candidly, especially cases where generated reports contain clinically serious findings absent from the reference.

7. Would strengthen: Releasing code, rule extraction scripts, prompts, and preference data construction details would substantially improve reproducibility.

---

> ### Author Response · Authors · 2026-06-18
>
> ### Additional Validation of Rule Extraction and Verifier Performance
> - We thank the reviewer for this valuable suggestion. Our primary goal is to evaluate the effectiveness of the extracted rules and verifier within the report generation framework. We would like to clarify that the rule extraction process is described in detail in the manuscript and was developed in consultation with clinicians (alongwith one of the authors is also a trained physician), who verified the clinical validity of the extracted concept–modifier relationships and rule templates. We agree that additional verifier-specific analysis would strengthen the paper. In the revised manuscript, we have included verifier performance metrics (in Appendix A.2.1) on a held-out validation set, including precision, recall, and F1-score. The scores are precision: 0.9863, recall:0.9703, and F1-score: 0.9816. This set of scores give us the idea that the final scores reflect the verifier influence properly. We also note that the current manuscript already contains analyses of verifier behavior through cross-institution (which in our case refers to the two datasets MIMIC-CXR-JPG and IUX-ray originated from MIT and Indiana University respectively) experiments and verifier sensitivity studies at varying verifier accuracies. These experiments provide evidence regarding the robustness and effectiveness of the verifier beyond the primary report-generation results.
>
> ### Effects of Rule-Guided Preference Curation and Verifier-Based Regularization
> - We thank the reviewer for this suggestion. We would like to clarify that the DPO baseline in our experiments is trained using the same rule-guided preference pairs as our proposed method. Therefore, the effect of rule-guided preference data curation is already captured, and the observed performance differences can be attributed to the additional symbolic verifier regularization introduced by our framework. In other words, the comparison between DPO and our method already corresponds to using identical preference data with and without the verifier-based objective. We will revise the manuscript to make this experimental setup more explicit. We agree that studying the robustness of the framework under noisy or randomly perturbed rules and investigating different strategies for setting λ are interesting directions we will add further ablations in the revised manuscript.
>
> | Method | B-1 | B-2 | B-3 | B-4 | RG-L | RadGraph-F1 | GREEN |
> |---|---|---|---|---|---|---|---|
> | + noisy-rules | 0.2552 ± 0.09 | 0.1525 ± 0.07 | 0.0826 ± 0.06 | 0.0454 ± 0.05 | 0.2509 ± 0.08 | 0.8695 ± 0.04 | 0.2906 ± 0.13 |
> | + fixed-lambda (=0.1) | 0.2652 ± 0.09 | 0.1537 ± 0.07 | 0.0826 ± 0.07 | 0.0404 ± 0.05 | 0.2465 ± 0.09 | 0.8605 ± 0.04 | 0.2783 ± 0.12 |
> | + fixed-lambda (=0.5) | 0.2375 ± 0.07 | 0.1334 ± 0.05 | 0.0696 ± 0.05 | 0.0332 ± 0.04 | 0.2209 ± 0.08 | 0.8581 ± 0.04 | 0.2688 ± 0.12 |
> | + SCOPE-RRG (Ours) | 0.2936 ± 0.10 | 0.1748 ± 0.08 | 0.0997 ± 0.07 | 0.0539 ± 0.06 | 0.2771 ± 0.10 | 0.8817 ± 0.03 | 0.3100 ± 0.13 |
>
> In Appendix A.4 we provide an elaborate empirical result that compares the new baselines with the existing. we analyze the impact of rule quality and verifier weighting through controlled ablations. Training with 50% noisy rules results in a performance degradation across both lexical and clinical metrics, demonstrating that the verifier benefits from meaningful clinical rule structures rather than simple entity overlap. Furthermore, replacing the adaptive $\lambda$ with fixed values ($\lambda=0.1$ and $\lambda=0.5$) consistently reduces performance, highlighting the importance of adaptive verifier regularization in effectively integrating the rule-based supervision.
>
> ### Optimization and Experimental Setup Details
> - We thank the reviewer for pointing out this important information. Our manuscript already contains the mentioned details in Appendix B.1, which provides the experimental setting for all the training required. The dataset preparation details are already provided in Appendices A.1,A.2, A.3. Additionally, we have added a description of the verifier-based loss and decoding details in Appendix B.2
>
> ### Statistical Validation and Robustness Analysis
> - We thank the reviewer for this suggestion. To better quantify the robustness of our results, we will include standard deviations as shown previously, we have added elaborate empirical results with standard deviations in Appendix A.4. We have revised the discussion to more carefully reflect the mixed behavior across metrics. We have replaced “consistent gains” with positive increment across majority lexical and semantic metrics, in the revised manuscript.
>
> ### Reproducibility Enhancements and Analysis
> -  We will also release the code, rule extraction scripts, prompts, and data-processing pipeline upon acceptance, and add a dedicated reproducibility statement to improve accessibility of these resources. Additionally, we add both positive and negative samples in App. A.3

---

> > ### Author Response · Authors · 2026-06-18
> >
> > ### Clinical Evaluation and Error Characterization
> > - We thank the reviewer for this suggestion and agree that larger-scale clinical evaluation would provide additional evidence of the effectiveness of the proposed approach. While a large blinded study involving multiple radiologists is not feasible given the costs associated for doing so (also preventive low resource setting labs to being handicapped and non-competitive at the global stage if these are mandatory expectations), we do provide both expert-based and LLM-based evaluations in the current manuscript. Specifically, we include a limited (but diverse)-scale human evaluation conducted by clinical experts, as well as an LLM-as-a-judge evaluation to assess report quality from complementary perspectives.
> >
> > In addition, we report clinically oriented automatic metrics such as RadGraph-F1 and GREEN (both specifically designed so that the field moves away from NLP metrics to clinically grounded metrics), which have been shown to correlate with clinical correctness and factual consistency. Our framework is also compared against strong baseline methods under identical experimental settings. We clarify these evaluation protocols in the revised manuscript and expand their discussion.

---

### Author Response · Authors · 2026-06-18

We sincerely, request for an extension of deadline. The volume of suggested experiments were high. The stipulated time of two weeks weren't enough given our current access to hardware. Consequently, we could not provide results for a few experiments suggested by Reviewer EPf1. We kindly request the AE to extend our rebuttal deadline by one week (Deadline: 26 Jun, 2026). Please, consider our proposition.

---

### Author Response · Authors · 2026-06-25

Dear Reviewers and AE,
We have added the additional empirical results and revised the manuscript accordingly. We would be eager to engage in further discussion regarding our work.
Thanks and Regards,